# Neural Conditional Probability for Uncertainty Quantification

**Vladimir R. Kostic**[1,2]    **Karim Lounici**[3]    **Grégoire Pacreau**[3]
**Giacomo Turri**[1]    **Pietro Novelli**[1]    **Massimiliano Pontil**[1,4]

[1]CSML, Istituto Italiano di Tecnologia    [2]University of Novi Sad
[3]CMAP-Ecole Polytechnique    [4]AI Centre, University College London

## Abstract

We introduce Neural Conditional Probability (NCP), an operator-theoretic approach to learning conditional distributions with a focus on statistical inference tasks. NCP can be used to build conditional confidence regions and extract key statistics such as conditional quantiles, mean, and covariance. It offers streamlined learning via a single unconditional training phase, allowing efficient inference without the need for retraining even when conditioning changes. By leveraging the approximation capabilities of neural networks, NCP efficiently handles a wide variety of complex probability distributions. We provide theoretical guarantees that ensure both optimization consistency and statistical accuracy. In experiments, we show that NCP with a 2-hidden-layer network matches or outperforms leading methods. This demonstrates that a a minimalistic architecture with a theoretically grounded loss can achieve competitive results, even in the face of more complex architectures.

## 1  Introduction

This paper studies the problem of estimating the conditional distribution associated with a pair of random variables, given a finite sample from their joint distribution. This problem is fundamental in machine learning, and instrumental for various purposes such as building prediction intervals, performing downstream analysis, visualizing data, and interpreting outcomes. This entails predicting the probability of an event given certain conditions or variables, which is a crucial task across various domains, ranging from finance (Markowitz, 1958) to medicine (Ray et al., 2017), to climate modeling (Harrington, 2017) and beyond. For instance, in finance, it is essential for risk assessment to estimate the probability of default given economic indicators. Similarly, in healthcare, predicting the likelihood of a disease, given patient symptoms, aids in diagnosis. In climate modeling, estimating the conditional probability of extreme weather events such as hurricanes or droughts, given specific climate indicators, helps in disaster preparedness and mitigation efforts.

According to Gao and Hastie (2022), there exist four main strategies to learn the conditional distribution. The first one relies on the Bayes formula for densities and proposes to apply non-parametric statistics to learn the joint and marginal densities separately. However, most of non-parametric techniques face a significant challenge known as the curse of dimensionality (Scott, 1991; Nagler and Czado, 2016). The second strategy, also known as Localization method, involves training a model unconditionally on reweighted samples, where weights are determined by their proximity to the desired conditioning point (Hall et al., 1999; Yu and Jones, 1998). These methods require retraining the model whenever the conditioning changes and may also suffer from the curse of dimensionality if the weighting strategy treats all covariates equally. The third strategy, known as Direct Learning of the conditional distribution involves finding the best linear approximation of the conditional density on a dictionary of base functions or a kernel space (Sugiyama et al., 2010; Li et al., 2007). The

performance of these methods relies crucially on the selection of bases and kernels. Again for high-dimensional settings, approaches that assign equal importance to all covariates may be less effective. Finally, the fourth strategy, known as Conditional Training, involves training models to estimate a target variable conditioned on certain covariates. This is typically based on partitioning the covariates space $\mathcal{X}$ into sets, followed by training models unconditionally within each partition (see Gao and Hastie, 2022; Winkler et al., 2020; Lu and Huang, 2020; Dhariwal and Nichol, 2021, and references therein). However, this strategy requires a large dataset to provide enough samples for each conditioning and is expensive as it requires training separate models for each conditioning input set, even though they stem from the same underlying joint distribution.

**Contributions** The principal contribution of this work is a different conditional probability approach that does not fall into any of the four aforementioned strategies. Rather than learning the conditional density directly, our method, called Neural Conditional Probability (NCP), aims to learn the *conditional expectation operator* $\mathsf{E}_{Y|X}$ associated to the random variables $X \in \mathcal{X}$ and $Y \in \mathcal{Y}$ based on data from their joint distribution. The operator is defined, for every measurable function $f : \mathcal{Y} \to \mathbb{R}$, as

$$[\mathsf{E}_{Y|X}f](x) := \mathbb{E}[f(Y) \,|\, X = x].$$

NCP is based on a principled loss, leveraging the connection between conditional expectation operators and deepCCA (Andrew et al., 2013) established in (Kostic et al., 2024), and can be used interchangeably to:

(a) retrieve the conditional density $p_{Y|X}$ with respect to marginal distributions of $X$ and $Y$;

(b) compute conditional statistics $\mathbb{E}[f(Y) \,|\, X]$ for arbitrary functions $f : \mathcal{Y} \to \mathbb{R}$, including conditional mean, variance, moments, and the conditional cumulative distribution function, thereby providing access to all conditional quantiles simultaneously;

(c) estimate the conditional probabilities $\mathbb{P}[Y \in B \,|\, X \in A]$ for arbitrary sets $B \subset \mathcal{Y}$ and $A \subset \mathcal{X}$ with theoretical non-asymptotic guarantees on accuracy, allowing us to easily construct conditional confidence regions.

Notably, our approach extracts statistics directly from the trained operator without retraining or resampling, and it is supported by both optimization consistency and statistical guarantees. In addition our experiments show that our approach matches or exceeds the performance of leading methods, even when using a basic a 2-hidden-layer network. This demonstrates the effectiveness of a minimalistic architecture combined with a theoretically grounded loss function.

**Paper organization** In Section 2 we review related work. Section 3 introduces the operator theoretic approach to model conditional expectation, while Section 4 discusses its training pipeline. In Section 5, we derive learning guarantees for NCP. Finally, Section 6 presents numerical experiments.

## 2 Related works

Non-parametric estimators are valuable for density and conditional density estimation as they don't rely on specific assumptions about the density being estimated. Kernel estimators, pioneered by Parzen (1962) and Rosenblatt (1956), are a widely used non-parametric density estimation method. Much effort has been dedicated to enhancing kernel estimation, focusing on aspects like bandwidth selection (Goldenshluger and Lepski, 2011), non-linear aggregation (Rigollet and Tsybakov, 2007), and computational efficiency (Langrené and Warin, 2020), as well as extending it to conditional densities (Bertin et al., 2014). A comprehensive review of kernel estimators and their variants is provided in (Silverman, 2017). See also (Tsybakov, 2009) for a statistical analysis of their performance. However, most of non-parametric techniques face a significant challenge known as the curse of dimensionality (Scott, 1991; Nagler and Czado, 2016), meaning that the required sample size for accurate estimation grows exponentially with the dimensionality of the data (Silverman, 2017). Additionally, the computational complexity also increases exponentially with dimensionality (Langrené and Warin, 2020).

Examples of localization methods include the work by Hall et al. (1999) for conditional CDF estimation using local logistic regression and locally adjusted Nadaraya-Watson estimation, as well as conditional quantiles estimation via local pinball loss minimization in (Yu and Jones, 1998). Examples of direct learning of the conditional distribution include (Sugiyama et al., 2010) via decomposition on

a dictionary of base functions. Similarly, Li et al. (2007) explores quantile regression in reproducing Hilbert kernel spaces.

Conditional training is a popular approach which was adopted in numerous works, as in the recent work by Gao and Hastie (2022) where a parametric exponential model for the conditional density $p_\theta(y|x)$ is trained using the Lindsey method within each bin of a partition of the space $\mathcal{X}$. This strategy has also been implemented in several prominent classes of generative models, including Normalizing Flow (NF) and Diffusion Models (DM) (Tabak and Vanden-Eijnden, 2010; Dinh et al., 2014; Rezende and Mohamed, 2015a; Sohl-Dickstein et al., 2015). These models work by mapping a simple probability distribution into a more complex one. Conditional training approaches for NF and DM have been developed in many works including (e.g. Winkler et al., 2020; Lu and Huang, 2020; Dhariwal and Nichol, 2021). In efforts to lower the computational burden of conditional diffusion models, an alternative approach used heuristic approximations applied directly to unconditional diffusion models on computer vision related tasks (see e.g. Song et al., 2023; Zhang et al., 2023). However, the effectiveness of these heuristics in accurately mimicking the true conditional distributions remains uncertain. Another crucial aspect of these classes of generative models is that while the probability distribution is modelled explicitly, the computation of any relevant statistic, say $\mathbb{E}[Y|X]$ is left as an implicit problem usually solved by sampling from $p_\theta(y|x)$ and then approximating $\mathbb{E}[Y|X]$ via simple Monte-Carlo integration. As expected, this approach quickly becomes problematic as the dimension of the output space $\mathcal{Y}$ becomes large.

Conformal Prediction (CP) is a popular model-agnostic framework for uncertainty quantification (Vovk et al., 1999). Conditional Conformal Prediction (CCP) was later developed to handle conditional dependencies between variables, allowing in principle for more accurate and reliable predictions (see Lei and Wasserman, 2014; Romano et al., 2019; Chernozhukov et al., 2021; Gibbs et al., 2023, and the references cited therein). However, (CP) and (CCP) are not without limitations. The construction of these guaranteed prediction regions need to be recomputed from scratch for each value of the confidence level parameter and of the conditioning for (CCP). In addition, the produced confidence regions tend to be conservative.

## 3   Operator approach to probability modeling

Consider a pair of random variables $X$ and $Y$ taking values in probability spaces $(\mathcal{X}, \Sigma_{\mathcal{X}}, \mu)$ and $(\mathcal{Y}, \Sigma_{\mathcal{Y}}, \nu)$, respectively, where $\mathcal{X}$ and $\mathcal{Y}$ are state spaces, $\Sigma_{\mathcal{X}}$ and $\Sigma_{\mathcal{Y}}$ are sigma algebras, and $\mu$ and $\nu$ are probability measures. Let $\rho$ be the joint probability measure of $(X, Y)$ from the product space $\mathcal{X} \times \mathcal{Y}$. We assume that $\rho$ is absolutely continuous w.r.t. to the product measure of its marginals, that is $\rho \ll \mu \times \nu$, and denote the corresponding density by $p = d\rho/d(\mu \times \nu)$, also called point-wise dependency in Tsai et al. (2020), so that $\rho(dx, dy) = p(x, y)\mu(dx)\nu(dy)$.

The principal goal of this paper is, given a dataset $\mathcal{D}_n := (x_i, y_i)_{i\in[n]}$ of observations of $(X, Y)$, to estimate the conditional probability measure

$$p(B \,|\, x) := \mathbb{P}[Y \in B \,|\, X = x], \quad x \in \mathcal{X}, B \in \Sigma_{\mathcal{Y}}. \tag{1}$$

Our approach is based on the simple fact that $p(B \,|\, x) = \mathbb{E}[\mathbb{1}_B(Y) \,|\, X = x]$, where $\mathbb{1}_B$ denotes the characteristic function of set $B$. More broadly we address the above problem by studying the conditional expectation operator $\mathsf{E}_{Y|X} \colon L^2_\nu(\mathcal{Y}) \to L^2_\mu(\mathcal{X})$, which is defined, for every $f \in L^2_\nu(\mathcal{Y})$ and $x \in \mathcal{X}$, as

$$[\mathsf{E}_{Y|X}f](x) := \mathbb{E}[f(Y) \,|\, X = x] = \int_{\mathcal{Y}} f(y)p(dy \,|\, x) = \int_{\mathcal{Y}} f(y)p(x, y)\nu(dy),$$

where $L^2_\mu(\mathcal{X})$ and $L^2_\nu(\mathcal{Y})$ denotes the Hilbert spaces of functions that are square integrable w.r.t. to $\mu$ and $\nu$, respectively. One readily verifies that $\|\mathsf{E}_{Y|X}\| = 1$ and $\mathsf{E}_{Y|X}\mathbb{1}_{\mathcal{Y}} = \mathbb{1}_{\mathcal{X}}$.

A prominent feature of the above operator is that its rank can reveal the independence of the random variables. That is, $X$ and $Y$ are independent random variables if and only if $\mathsf{E}_{Y|X}$ is a rank one operator, in which case we have that $\mathsf{E}_{Y|X} = \mathbb{1}_{\mathcal{X}} \otimes \mathbb{1}_{\mathcal{Y}}$. It is thus useful to consider the deflated operator $\mathsf{D}_{Y|X} = \mathsf{E}_{Y|X} - \mathbb{1}_{\mathcal{X}} \otimes \mathbb{1}_{\mathcal{Y}} \colon L^2_\nu(\mathcal{Y}) \to L^2_\mu(\mathcal{X})$, for which we have that

$$[\mathsf{E}_{Y|X}f](x) = \mathbb{E}[f(Y)] + [\mathsf{D}_{Y|X}f](x), \quad f \in L^2_\nu(\mathcal{Y}). \tag{2}$$

For dependent random variables, the deflated operator is nonzero. In many important situations, such as when the conditional probability distribution is a.e. absolutely continuous w.r.t. to the target

measure, that is $p(\cdot \mid x) \ll \nu$ for $\mu$-a.e. $x \in \mathcal{X}$, the operator $\mathsf{E}_{Y|X}$ is compact, and, hence, we can write the SVD of $\mathsf{E}_{Y|X}$ and $\mathsf{D}_{Y|X}$ respectively as

$$\mathsf{E}_{Y|X} = \sum_{i=0}^{\infty} \sigma_i^\star \, u_i^\star \otimes v_i^\star, \quad \text{and} \quad \mathsf{D}_{Y|X} = \sum_{i=1}^{\infty} \sigma_i^\star \, u_i^\star \otimes v_i^\star, \tag{3}$$

where the left $(u_i^\star)_{i \in \mathbb{N}}$ and right $(v_i^\star)_{i \in \mathbb{N}}$ singular functions form complete orthonormal systems of $L_\mu^2(\mathcal{X})$ and $L_\nu^2(\mathcal{Y})$, respectively. Notice that the only difference in the SVD of $\mathsf{E}_{Y|X}$ and $\mathsf{D}_{Y|X}$ is the extra leading singular triplet $(\sigma_0^\star, u_0^\star, v_0^\star) = (1, \mathbb{1}_\mu, \mathbb{1}_\nu)$ of $\mathsf{E}_{Y|X}$. In terms of densities, the SVD of $\mathsf{E}_{Y|X}$ leads to the characterization

$$p(x, y) = \sum_{i=0}^{\infty} \sigma_i^\star u_i^\star(x) \, v_i^\star(y) = 1 + \sum_{i=1}^{\infty} \sigma_i^\star u_i^\star(x) \, v_i^\star(y).$$

The mild assumption that $\mathsf{E}_{Y|X}$ is a compact operator allows one to approximate it arbitrarily well with a (large enough) finite rank (empirical) operator. Choosing the operator norm as the measure of approximation error and appealing to the Eckart-Young-Mirsky Theorem (see Theorem 3 in Appendix B.1) one concludes that the best approximation is given by the truncated SVD, that is for every $d \in \mathbb{N}$,

$$\mathsf{D}_{Y|X} \approx [\![\mathsf{D}_{Y|X}]\!]_d := \sum_{i=1}^{d} \sigma_i^\star u_i^\star \otimes v_i^\star, \quad \text{and} \quad [\![\mathsf{D}_{Y|X}]\!]_d \in \arg\min_{\mathrm{rank}(A) \le d} \|\mathsf{D}_{Y|X} - A\|, \tag{4}$$

where the minimum is given by $\sigma_d^\star$, and the minimizer is unique whenever $\sigma_{d+1}^\star < \sigma_d^\star$. This leads to the approximation of the joint density w.r.t. marginals $p(x, y) \approx 1 + \sum_{i=1}^{d} \sigma_i^\star u_i^\star(x) \, v_i^\star(y)$, so that

$$\mathbb{E}[f(Y) \mid X = x] \approx \mathbb{E}[f(Y)] + \sum_{i=1}^{d} \sigma_i^\star u_i^\star(x) \mathbb{E}[f(Y) \, v_i^\star(Y)], \tag{5}$$

which in particular, choosing $f = \mathbb{1}_B$, gives

$$\mathbb{P}[Y \in B \mid X = x] \approx \mathbb{P}[Y \in B] + \sum_{i=1}^{d} \sigma_i^\star u_i^\star(x) \, \mathbb{E}[v_i^\star(Y) \mathbb{1}_B(Y)].$$

Moreover, we have that

$$\mathbb{P}[Y \in B \mid X \in A] = \frac{\langle \mathbb{1}_A, \mathsf{E}_{Y|X} \mathbb{1}_B \rangle}{\mathbb{P}[X \in A]} \approx \mathbb{P}[Y \in B] + \sum_{i=1}^{d} \sigma_i^\star \frac{\mathbb{E}[u_i^\star(X) \mathbb{1}_A(X)]}{\mathbb{P}[X \in A]} \mathbb{E}[v_i^\star(Y) \mathbb{1}_B(Y)],$$

for which the approximation error is bounded in the following lemma.

**Lemma 1** (Approximation bound). *For any $A \in \Sigma_{\mathcal{X}}$ such that $\mathbb{P}[X \in A] > 0$ and any $B \in \Sigma_{\mathcal{Y}}$,*

$$\left| \mathbb{P}[Y \in B \mid X \in A] - \mathbb{P}[Y \in B] - \frac{\langle \mathbb{1}_A, [\![\mathsf{D}_{Y|X}]\!]_d \mathbb{1}_B \rangle}{\mathbb{P}[X \in A]} \right| \le \sigma_{d+1}^\star \sqrt{\frac{\mathbb{P}[Y \in B]}{\mathbb{P}[X \in A]}}. \tag{6}$$

**Neural network model** Inspired by the above observations, to build the NCP model, we will parameterize the truncated SVD of the conditional expectation operator and then learn it. Specifically, we introduce two parameterized embeddings $u^\theta \colon \mathcal{X} \to \mathbb{R}^d$ and $v^\theta \colon \mathcal{Y} \to \mathbb{R}^d$, and the singular values parameterized by $w^\theta \in \mathbb{R}^d$, respectively given by

$$u^\theta(x) := [u_1^\theta(x) \ \ldots \ u_d^\theta(x)]^\top, \ \ v^\theta(y) := [v_1^\theta(y) \ \ldots \ v_d^\theta(y)]^\top, \ \text{and} \ \sigma^\theta := [e^{-(w_1^\theta)^2}, \ldots, e^{-(w_d^\theta)^2}]^\top,$$

where the parameter $\theta$ takes values in a prescribed set $\Theta$.

We then aim to learn the joint density function $p(x, y)$ in the (separable) form

$$p_\theta(x, y) := 1 + \sum_{i \in [d]} \sigma_i^\theta u_i^\theta(x) \, v_i^\theta(y) = 1 + \langle \sigma^\theta \odot u^\theta(x), v^\theta(y) \rangle,$$

where $\odot$ denotes element-wise product. One of the prominent losses considered for the task of learning $p \in \mathcal{L}_{\mu \times \nu}^2$ is *the least squares density ratio* loss $\mathbb{E}_{\mu \times \nu}(p - p_\theta)^2 - \mathbb{E}_\rho p = \mathbb{E}_{\mu \times \nu} p_\theta^2 - 2\mathbb{E}_\rho p_\theta$, c.f. Tsai et al. (2020), also considered by HaoChen et al. (2022) in the specific context of augmentation graph in self-supervised deep learning, linked to kernel embeddings (Wang et al., 2022), and rediscovered and tested on DeepCCA tasks by Wells et al. (2024). Here, following the operator perspective, we use the characterization (4) of the optimal finite rank model to propose a new loss that: (1) excludes the known feature from the learning process, and (2) introduces a penalty term to enforce orthonormality of the basis functions. More precisely, our loss $\mathcal{L}_\gamma(\theta) := \mathcal{L}(\theta) + \gamma \mathcal{R}(\theta)$ is composed of two terms. The first one

$$\mathcal{L}(\theta) := \mathbb{E}_{\mu \times \nu} p_\theta^2 - 2\mathbb{E}_\rho p_\theta + [\mathbb{E}_{\mu \times \nu} p_\theta]^2 - \mathbb{E}_\mu [\mathbb{E}_\nu p_\theta]^2 - \mathbb{E}_\nu [\mathbb{E}_\mu p_\theta]^2 + 2\mathbb{E}_{\mu \times \nu} p_\theta \tag{7}$$

is equivalent to solving (4) with $A = \sum_{i=1}^{d} \sigma_i^\theta \, [u_i^\theta - \mathbb{E}_\mu u_i^\theta] \otimes [v_i^\theta - \mathbb{E}_\nu v_i^\theta]$ and can be written in terms of correlations between features. Namely, denoting the covariance and variance matrices by

$$\mathrm{Cov}[z, z'] := \mathbb{E}[(z - \mathbb{E}[z])(z' - \mathbb{E}[z'])^\top] \quad \text{and} \quad \mathrm{Var}[z] := \mathbb{E}[(z - \mathbb{E}[z])(z - \mathbb{E}[z])^\top], \quad (8)$$

and abbreviating $u^\theta := u^\theta(X)$ and $v^\theta := v^\theta(Y)$ for simplicity, we can write

$$\mathcal{L}(\theta) := \mathrm{tr}\left(\mathrm{Var}[\sqrt{\sigma^\theta} \odot u^\theta]\,\mathrm{Var}[\sqrt{\sigma^\theta} \odot v^\theta] - 2\,\mathrm{Cov}[\sqrt{\sigma^\theta} \odot u^\theta, \sqrt{\sigma^\theta} \odot v^\theta]\right). \quad (9)$$

If $p = p_\theta$ for some $\theta \in \Theta$, then the optimal loss is the $\chi^2$-divergence $\mathcal{L}(\theta) = D_{\chi^2}(\rho \mid \mu \times \nu) = -\sum_{i \geq 1} \sigma_i^{\star 2}$ and, as we show below, $\mathcal{L}(\theta)$ measures how well $p_\theta(x, y) - 1$ approximates $\sum_{i \in [d]} \sigma_i^\star u_i^\star(x)\, v_i^\star(y)$. However, in order to obtain a useful probability model, it is of paramount importance to *align* the metric in the latent spaces with the metrics in the data-spaces $L_\mu^2(\mathcal{X})$ and $L_\nu^2(\mathcal{Y})$. For different reasons, a similar phenomenon has been observed in Kostic et al. (2024) where dynamical systems are learned via transfer operators. In our setting, this leads to the second term of the loss that measures how well features $u^\theta$ and $v^\theta$ span relevant subspaces in $L_\mu^2(\mathcal{X})$ and $L_\nu^2(\mathcal{Y})$, respectively. Namely, aiming $\mathbb{E}[u_i^\star(X)u_j^\star(X)] = \mathbb{E}[v_i^\star(Y)v_j^\star(Y)] = \mathbb{1}_{\{i=j\}}$, $i, j \in \{0, 1, \ldots, d\}$ leads to

$$\mathcal{R}(\theta) := \|\mathbb{E}[u^\theta(X)u^\theta(X)^\top] - I\|_F^2 + \|\mathbb{E}[v^\theta(Y)v^\theta(Y)^\top] - I\|_F^2 + 2\|\mathbb{E}[u^\theta(X)]\|^2 + 2\|\mathbb{E}[v^\theta(Y)]\|^2. \quad (10)$$

We now state our main result on the properties of the loss $\mathcal{L}_\gamma$, which extends the result in Wells et al. (2024) to infinite-dimensional operators and guarantees the uniqueness of the optimum due to $\mathcal{R}$.

**Theorem 1.** *Let* $\mathsf{E}_{Y|X} : L_\nu^2(\mathcal{Y}) \to L_\mu^2(\mathcal{X})$ *be a compact operator and* $\mathsf{D}_{Y|X} = \sum_{i=1}^\infty \sigma_i^\star u_i^\star \otimes v_i^\star$ *be the SVD of its deflated version. If* $u_i^\theta \in L_\mu^2(\mathcal{X})$ *and* $v_i^\theta \in L_\nu^2(\mathcal{Y})$, *for all* $\theta \in \Theta$ *and* $i \in [d]$, *then for every* $\theta \in \Theta$, $\mathcal{L}_\gamma(\theta) \geq -\sum_{i \in [d]} \sigma_i^{\star 2}$. *Moreover, if* $\gamma > 0$ *and* $\sigma_d^\star > \sigma_{d+1}^\star$, *then the equality holds if and only if* $(\sigma_i^\theta, u_i^\theta, v_i^\theta)$ *equals* $(\sigma_i^\star, u_i^\star, v_i^\star)$ $\rho$-a.e., *up to unitary transform of singular spaces.*

We provide the proof in Appendix B.3. In the following section, we show how to learn these canonical features from data and construct approximations of the conditional probability measure.

**Comparison to previous methods**  NCP does not fall into any of the four categories defined by Gao and Hastie (2022), as it does not aim to learn conditional density of $Y|X$ directly. Instead, NCP focuses on learning the operator mapping $L_\nu^2(\mathcal{Y}) \to L_\mu^2(\mathcal{X})$, from which all relevant task-specific statistics can be derived without requiring retraining. This approach effectively integrates with deep representation learning to create a latent space adapted to $p(y|x)$. As a result, NCP efficiently captures the intrinsic dimension of the data, which is supported by our theoretical guarantees that depend solely on the latent space dimension (Theorem 2). In contrast, strategies designed for learning density often encounter significant limitations, such as the curse of dimensionality, potential substantial misrepresentation errors when the pre-specified function dictionary misaligns with the true distribution $p(y|x)$, and high computational complexity due to the need for retraining. Experiments confirm NCP's capability to learn representations tailored to a wide range of data types—including manifolds, graphs, and high-dimensional distributions—without relying on predefined dictionaries. This flexibility allows NCP to outperform popular aforementioned methods.

## 4  Training the NCP inference method

In this section, we discuss how to train the model. Given a training dataset $\mathcal{D}_n = (x_i, y_i)_{i \in [n]}$ and networks $(u^\theta, v^\theta, \sigma^\theta)$, we consider the empirical loss $\widehat{\mathcal{L}}_\gamma(\theta) := \widehat{\mathcal{L}}(\theta) + \gamma\widehat{\mathcal{R}}(\theta)$, where we replaced (9) and (10) by their empirical versions. In order to guarantee the unbiased estimation, as we show within the proof of Theorem 1, two terms of our loss can be written using two independent samples $(X, Y)$ and $(X', Y')$ from $\rho$ as $\mathcal{L}(\theta) = \mathbb{E}[L(u^\theta(X) - \mathbb{E}u^\theta(X), u^\theta(X') - \mathbb{E}u^\theta(X'), v^\theta(Y) - \mathbb{E}v^\theta(Y), v^\theta(Y') - \mathbb{E}v^\theta(Y'), \sigma^\theta)]$ and $\mathcal{R}(\theta) = \mathbb{E}[R(u^\theta(X), u^\theta(X'), v^\theta(Y), v^\theta(Y'))]$, where the loss functionals $L$ and $R$ are defined for $u, u', v, v' \in \mathbb{R}^d$ and $s \in [0, 1]^d$ as

$$L(u, u', v, v', s) := \tfrac{1}{2}\left(u^\top \mathrm{diag}(s)v'\right)^2 + \tfrac{1}{2}\left(v^\top \mathrm{diag}(s)u'\right)^2 - u^\top \mathrm{diag}(s)v' - v^\top \mathrm{diag}(s)u', \quad (11)$$

$$R(u, u', v, v') := (u^\top u')^2 - (u - u')^\top(u - u') + (v^\top v')^2 - (v - v')^\top(v - v') + 2d. \quad (12)$$

Therefore, at every epoch we take two independent batches $\mathcal{D}_n^1$ and $\mathcal{D}_n^2$ of equal size from $\mathcal{D}_n$, leading to Algorithm 1. See Appendix A.1 for the full discussion, and Appendix A.2, where we also provide in Figure 4 an example of learning dynamics.

---

**Algorithm 1** Separable density learning procedure

---

**Require:** training data $(X_{\text{train}}, Y_{\text{train}})$
   train $u^\theta$, $\sigma^\theta$ and $v^\theta$ using the NCP loss
   Center and scale $X_{\text{train}}$ and $Y_{\text{train}}$
   **for** each epoch **do**
      From $(X_{\text{train}}, Y_{\text{train}})$ pick two random batches $(X_{\text{train}}, Y_{\text{train}})$ and $(X'_{\text{train}}, Y'_{\text{train}})$
      Evaluate: $U \leftarrow u^\theta(X_{\text{train}})$, $U' \leftarrow u^\theta(X'_{\text{train}})$, $V \leftarrow v^\theta(Y_{\text{train}})$, $V' \leftarrow v^\theta(Y'_{\text{train}})$
      Compute $\widehat{\mathcal{L}}(\theta)$ as an unbiased estimate using (9) or (11)
      Compute $\widehat{\mathcal{R}}(\theta)$ as an unbiased estimate using (10) or (12)
      Compute NCP loss $\widehat{\mathcal{L}}_\gamma(\theta) := \widehat{\mathcal{L}}(\theta) + \gamma\widehat{\mathcal{R}}(\theta)$ and back-propagate
   **end for**

---

**Practical guidelines for training** In the following, we briefly report a few aspects to be kept in mind when using the NCP in practice, referring the reader to Appendix A for further details. The computational complexity of loss estimation presents three distinct methodological approaches. The first method utilizes unbiased estimation via covariance calculations in (9) and (10), achieving a computational complexity of $\mathcal{O}(nd^2)$ for a batch size $n$. An alternative approach employing U-statistics with (11) and (12) requires $\mathcal{O}(n^2 d)$ operations per iteration, offering the estimation of the same precision. A third method involves batch averaging of (11) and (12), reducing computational complexity to $\mathcal{O}(nd)$, which enables seamless integration with contemporary deep learning frameworks, albeit potentially compromising training robustness through less accurate 4th-order moment estimations. Method selection remains contingent upon the specific problem's computational and statistical constraints. Further, the size of latent dimension $d$, as indicated by Theorem 1 relates to the problem's "difficulty" in the sense of smoothness of joint density w.r.t. its marginals. Lastly, after the training, an additional post-processing may be applied to ensure the orthogonality of features $u^\theta$ and $v^\theta$ and improve statistical accuracy of the learned model.

**Performing inference with the trained NCP model** We now explain how to extract important statistical objects from the trained model $(\widehat{u}^\theta, \widehat{v}^\theta, \sigma^\theta)$. To this end, define the empirical operator

$$\widehat{\mathsf{D}}^\theta_{Y|X}: L^2_\nu(\mathcal{Y}) \to L^2_\mu(\mathcal{X}) \quad [\widehat{\mathsf{D}}^\theta_{Y|X} f](x) := \sum_{i\in[d]} \sigma^\theta_i \widehat{u}^\theta_i(x) \widehat{\mathsf{E}}_y[\widehat{v}^\theta_i f], \quad f \in L^2_\nu(\mathcal{Y}),\, x \in \mathcal{X}, \quad (13)$$

where $\widehat{\mathsf{E}}_y[\widehat{v}^\theta_i f] := \frac{1}{n}\sum_{j\in[n]} \widehat{v}^\theta_i(y_j) f(y_j)$. Then, *without any retraining nor simulation*, we can compute the following statistics:

▶ Conditional Expectation: $[\widehat{\mathsf{E}}^\theta_{Y|X} f](x) := \widehat{\mathsf{E}}_y f + [\widehat{\mathsf{D}}^\theta_{Y|X} f](x)$, $f \in L^2_\nu(\mathcal{Y}), x \in \mathcal{X}$.

▶ Conditional moments of order $\alpha \geq 1$: apply previous formula to $f(u) = u^\alpha$.

▶ Conditional covariance: $\widehat{\mathrm{Cov}}^\theta(Y|X) := \widehat{\mathsf{E}}^\theta_{Y|X}[YY^\top] - \widehat{\mathsf{E}}^\theta_{Y|X}[Y]\widehat{\mathsf{E}}^\theta_{Y|X}[Y^\top]$.

▶ Conditional probabilities: apply the above conditional expectation formula with $f(y) = \mathbb{1}_B(y)$, that is, $\widehat{p}_y(B) = \widehat{\mathsf{E}}_y[\mathbb{1}_B]$ and $\widehat{p}_\theta(B\,|\,x) = \widehat{p}_y(B) + \sum_{i\in[d]} \sigma^\theta_i \widehat{u}^\theta_i(x) \widehat{\mathsf{E}}_y[\widehat{v}^\theta_i \mathbb{1}_B]$, $B \in \Sigma_\mathcal{Y}, x \in \mathcal{X}$. Then, integrating over an arbitrary set $A \in \Sigma_\mathcal{X}$ we get

$$\widehat{p}_\theta(B\,|\,A) := \widehat{p}_y(B) + \sum_{i\in[d]} \sigma^\theta_i \frac{\widehat{\mathsf{E}}_x[\widehat{u}^\theta_i \mathbb{1}_A]}{\widehat{\mathsf{E}}_x[\mathbb{1}_A]} \widehat{\mathsf{E}}_y[\widehat{v}^\theta_i \mathbb{1}_B]. \quad (14)$$

▶ Conditional quantiles: for scalar output $Y$, the conditional CDF $\widehat{F}_{Y|X\in A}(t)$ is obtained by taking $B = (-\infty, t]$, and in Algorithm 3 in Appendix C we show how to extract quantiles from it.

## 5 Statistical guarantees

We introduce some standard assumptions needed to state our theoretical learning guarantees. To that end, for any $A \in \Sigma_\mathcal{X}$ and $B \in \Sigma_\mathcal{Y}$ we define important constants, followed by the main assumption,

$$\varphi_X(A) := 1 \vee \sqrt{\frac{1 - \mathbb{P}[X \in A]}{\mathbb{P}[X \in A]}} \quad \text{and} \quad \varphi_Y(B) := 1 \vee \sqrt{\frac{1 - \mathbb{P}[Y \in B]}{\mathbb{P}[Y \in B]}}.$$

**Assumption 1.** *There exists finite absolute constants $c_u, c_v > 1$ such that for any $\theta \in \Theta$*

$$\operatorname*{ess\,sup}_{x \sim \mu} \|u^\theta(x)\|_{l_\infty} \le c_u, \quad \operatorname*{ess\,sup}_{y \sim \nu} \|v^\theta(y)\|_{l_\infty} \le c_v.$$

Next, we set $\sigma_\theta^2(X) := \operatorname{Var}(\|u^\theta(X) - \mathbb{E}[u^\theta(X)]\|_{l_2})$, $\sigma_\theta^2(Y) := \operatorname{Var}(\|v^\theta(Y) - \mathbb{E}[v^\theta(Y)]\|_{l_2})$ and

$$\epsilon_n(\delta) := C\left( (c_u \vee c_v) \frac{\sqrt{d}\,\log(e\delta^{-1})}{n} + (\sigma_\theta(X) \vee \sigma_\theta(Y)) \sqrt{\frac{\log(e\delta^{-1})}{n}} \right), \quad \bar{\epsilon}_n(\delta) := 2\sqrt{2\frac{\log 2\delta^{-1}}{n}}, \tag{15}$$

for some large enough absolute constant $C > 0$.

**Remark 1.** *It follows easily from Assumption 1 that $\sigma_\theta^2(X) \le c_u^2 d$ and $\sigma_\theta^2(Y) \le c_v^2 d$. Consequently, assuming that $n \ge (c_u \vee c_v)d$, then $\epsilon_n(\delta) \lesssim (c_u \vee c_v)\sqrt{d}[\sqrt{\log(e\delta^{-1})/n} \vee (\log(e\delta^{-1})/n)]$.*

Finally, for a given parameter $\theta \in \Theta$ and $\delta \in (0,1)$, let us denote

$$\mathcal{E}_\theta := \max\{\|[\![\mathrm{D}_{Y|X}]\!]_d - U_\theta S_\theta V_\theta^*\|, \|U_\theta^* U_\theta - I\|, \|U_\theta^* \mathbb{1}_\mathcal{X}\|, \|V_\theta^* V_\theta - I\|, \|V_\theta^* \mathbb{1}_\mathcal{Y}\|\}, \quad \text{and} \tag{16}$$

$$\psi_n(\delta) := \sigma_{d+1}^\star + \mathcal{E}_\theta + 2\sqrt{1 + \mathcal{E}_\theta}(\mathcal{E}_\theta + \varepsilon_n(\delta)) + [\varepsilon_n(\delta)]^2. \tag{17}$$

In the following result, we prove that NCP model approximates well the conditional probability distribution w.h.p. whenever the empirical loss $\widehat{\mathcal{L}}_\gamma(\theta)$ is well minimized.

**Theorem 2.** *Let Assumption 1 be satisfied, and in addition assume that*

$$\mathbb{P}(X \in A) \bigwedge \mathbb{P}(Y \in B) \ge \bar{\epsilon}_n(\delta/3) \quad \text{and} \quad n \ge (c_u \vee c_v)^2 d\sqrt{8\log(6\delta^{-1})}\,[\varphi_X(A) \vee \varphi_Y(B)]. \tag{18}$$

*Then for every $A \in \Sigma_\mathcal{X} \setminus \{\mathcal{X}\}$ and $B \in \Sigma_\mathcal{Y} \setminus \{\mathcal{Y}\}$*

$$\left| \frac{\mathbb{P}[Y \in B \mid X \in A]}{\mathbb{P}[Y \in B]} - \frac{\widehat{p}_\theta(B \mid A)}{\widehat{p}_y(B)} \right| \le \frac{4\psi_n(\delta/3) + [1 + \psi_n(\delta/3)]\,[2\varphi_X(A) + 4\varphi_Y(B)]\,\bar{\epsilon}_n(\delta/3)}{\sqrt{\mathbb{P}[X \in A]\mathbb{P}[Y \in B]}}, \tag{19}$$

*and*

$$\left| \frac{\mathbb{P}[Y \in B \mid X \in A] - \widehat{p}_\theta(B \mid A)}{\mathbb{P}[Y \in B]} \right| \le \varphi_Y(B)\bar{\epsilon}_n(\delta/3) + \frac{2(1 + \psi_n(\delta/3))\varphi_X(A)\bar{\epsilon}_n(\delta/3) + \psi_n(\delta/3)}{\sqrt{\mathbb{P}[X \in A]\mathbb{P}[Y \in B]}} \tag{20}$$

*hold with probability at least $1 - \delta$ w.r.t. iid draw of the dataset $\mathcal{D}_n = (x_j, y_j)_{j \in [n]}$ from $\rho$.*

**Remark 2.** *In Appendix B.5, we prove a similar result under a less restrictive sub-Gaussian assumption on the singular functions $u^\theta(X)$ and $v^\theta(Y)$.*

**Discussion** The rate $\psi_n(\delta)$ in (17) is pivotal for the efficacy of our method. If we appropriately choose the latent space dimension $d$ to ensure accurate approximation ($\sigma_{d+1}^\star \ll 1$), achieve successful training ($\mathcal{E}_\theta \approx \sigma_{d+1}^\star$), and secure a large enough sample size ($\varepsilon_n(\delta) \ll 1$), Theorem 2 provides assurance of accurate prediction of conditional probabilities. Indeed, (20) guarantees (up to a logarithmic factor)

$$\mathbb{P}[Y \in B \mid X \in A] - \widehat{p}_\theta(B \mid A) = O_\mathbb{P}\left( \frac{1}{\sqrt{n}} + \sqrt{\frac{\mathbb{P}[Y \in B]}{\mathbb{P}[X \in A]}}\left( \sigma_{d+1}^\star + \mathcal{E}_\theta + \sqrt{d/n} + \varphi_X(A)/\sqrt{n} \right) \right),$$

Note the inclusion of the term $\sqrt{\mathbb{P}[X \in A]}$ in the denominator of the last term on the right-hand side, along with $\varphi_X(A)$. This indicates a decrease in the accuracy of conditional probability estimates for rarely encountered event $A$, aligning with intuition and with a known finite-sample impossibility result Lei and Wasserman (2014, Lemma 1) for conditional confidence regions when $A$ is reduced to any nonatomic point of the distribution (i.e. $A = \{x\}$ with $\mathbb{P}[X = x] = 0$). For rare events, a larger sample size $n$ and a higher-dimensional latent space characterized by $d$ are necessary for accurate estimation of conditional probabilities.

We propose next a non-asymptotic estimation guarantee for the conditional CDF of $Y|X$ when $Y$ is a scalar output. This result ensures in particular that accurate estimation of the true quantiles is possible with our method. Fix $t \in \mathbb{R}$ and consider the set $B_t = (-\infty, t]$ meaning that $\mathbb{P}[Y \in B_t | X \in A] = F_{Y|X \in A}(t)$ and $\mathbb{P}[Y \in B_t] = F_Y(t)$. We define similarly for the NCP estimator of the conditional CDF $\widehat{F}_{Y|X \in A}(t) = \widehat{p}_\theta(B_t \mid A)$. The result follows from applying (20) to the set $B_t$.

Table 1: Mean and standard deviation of Kolmogorov-Smirnov distance of estimated CDF from the truth averaged over 10 repetitions with $n = 10^5$ (best method in red, second best in bold black).

| Model | LinearGaussian | EconDensity | ArmaJump | SkewNormal | GaussianMixture | LGGMD |
|---|---|---|---|---|---|---|
| NCP - W | $\mathbf{0.010} \pm \mathbf{0.000}$ | $0.005 \pm 0.001$ | $0.010 \pm 0.002$ | $0.008 \pm 0.001$ | $0.015 \pm 0.004$ | $0.047 \pm 0.005$ |
| DDPM | $0.410 \pm 0.340$ | $0.236 \pm 0.217$ | $0.338 \pm 0.317$ | $0.250 \pm 0.224$ | $0.404 \pm 0.242$ | $0.405 \pm 0.218$ |
| NF | $0.008 \pm 0.006$ | $\mathbf{0.006} \pm \mathbf{0.003}$ | $0.143 \pm 0.010$ | $0.032 \pm 0.002$ | $0.107 \pm 0.003$ | $0.254 \pm 0.004$ |
| KMN | $0.601 \pm 0.004$ | $0.362 \pm 0.017$ | $0.487 \pm 0.004$ | $0.381 \pm 0.009$ | $0.309 \pm 0.001$ | $0.224 \pm 0.005$ |
| MDN | $0.225 \pm 0.013$ | $0.048 \pm 0.001$ | $0.163 \pm 0.018$ | $0.087 \pm 0.001$ | $0.129 \pm 0.007$ | $0.176 \pm 0.013$ |
| LSCDE | $0.420 \pm 0.001$ | $0.118 \pm 0.002$ | $0.247 \pm 0.001$ | $0.107 \pm 0.001$ | $0.202 \pm 0.001$ | $0.268 \pm 0.024$ |
| CKDE | $0.120 \pm 0.000$ | $0.010 \pm 0.001$ | $0.072 \pm 0.001$ | $\mathbf{0.023} \pm \mathbf{0.001}$ | $0.048 \pm 0.001$ | $0.230 \pm 0.014$ |
| NNKCDE | $0.047 \pm 0.003$ | $0.036 \pm 0.003$ | $\mathbf{0.030} \pm \mathbf{0.004}$ | $0.030 \pm 0.002$ | $0.035 \pm 0.002$ | $0.183 \pm 0.006$ |
| RFCDE | $0.128 \pm 0.007$ | $0.141 \pm 0.009$ | $0.133 \pm 0.015$ | $0.142 \pm 0.012$ | $0.130 \pm 0.012$ | $0.121 \pm 0.006$ |
| FC | $0.095 \pm 0.005$ | $0.011 \pm 0.001$ | $0.033 \pm 0.002$ | $0.035 \pm 0.007$ | $\mathbf{0.016} \pm \mathbf{0.001}$ | $0.047 \pm 0.003$ |
| LCDE | $0.108 \pm 0.001$ | $0.026 \pm 0.001$ | $0.113 \pm 0.002$ | $0.075 \pm 0.006$ | $0.035 \pm 0.001$ | $0.124 \pm 0.002$ |

**Corollary 1.** *Let the Assumptions of Theorem 2 be satisfied. Then for any $t \in \mathbb{R}$ and $\delta \in (0, 1)$, it holds with probability at least $1 - \delta$ that*

$$|\widehat{F}_{Y|X \in A}(t) - F_{Y|X \in A}(t)| \leq \sqrt{F_Y(t)(1 - F_Y(t))}\bar{\epsilon}_n(\delta/3)$$
$$+ \sqrt{\frac{F_Y(t)}{\mathbb{P}[X \in A]}} \left( \sigma_{d+1}^\star + 2\sqrt{2}\mathcal{E}_\theta + (2\sqrt{2}+1)\epsilon_n(\delta/3) + 4\varphi_X(A)\bar{\epsilon}_n(\delta/3) \right). \quad (21)$$

An important application of Corollary 1 lies in uncertainty quantification when output $Y$ is a scalar. Indeed, for any $\alpha \in (0, 1/2)$, we can scan the empirical conditional CDF $\widehat{F}_{Y|X \in A}$ for values $t_\alpha < t'_\alpha$ such that $\widehat{F}_{Y|X \in A}(t'_\alpha) - \widehat{F}_{Y|X \in A}(t_\alpha) = 1 - \alpha$ and $t'_\alpha - t_\alpha$ is minimal. That way we define a non-asymptotic conditional confidence interval $\widehat{B}_\alpha := (t_\alpha, t'_\alpha]$ with approximate coverage $1 - \alpha$. More precisely we deduce from Corollary 1 that

$$|\mathbb{P}[Y \in \widehat{B}_\alpha \,|\, X \in A] - (1 - \alpha)| \leq \frac{1}{2}\bar{\epsilon}_n(\delta/6)$$
$$+ \sqrt{\frac{1}{\mathbb{P}[X \in A]}} \left( \sigma_{d+1}^\star + 2\sqrt{2}\mathcal{E}_\theta + (2\sqrt{2}+1)\epsilon_n(\delta/6) + 4\varphi_X(A)\bar{\epsilon}_n(\delta/6) \right). \quad (22)$$

In App B.6, we derive statistical guarantees for the conditional expectation and covariance of $Y$.

## 6 Experiments

**Conditional density estimation** We applied our NCP method to a benchmark of several conditional density models including those of Rothfuss et al. (2019); Gao and Hastie (2022). See Appendix C.1 for the complete description of the data models and the complete list of compared methods in Tab. 2 with references. We also plotted several conditional CDF along with our NCP estimators in Fig. 6. To assess the performance of each method, we use Kolmogorov-Smirnov (KS) distance between the estimated and the true conditional CDFs. We test each method on nineteen different conditional values uniformly sampled between the 5%- and 95%-percentile of $p(x)$ and computed the averaged performance over all the used conditioning values. In Tab. 1, we report mean performance (KS distance $\pm$ std) computed over 10 repetitions, each with a different seed. NCP with whitening (NCP–W) outperforms all other methods on 4 datasets, ties with FlexCode (FC) on 1 dataset, and ranks a close second on another one behind NF. These experiments underscore NCP's consistent performance. We also refer to Tab. 3 in App C.1 for an ablation study on post-treatments for NCP.

**Confidence regions** Our goal is to estimate conditional confidence intervals for two different data models (Laplace and Cauchy). We investigate the performance of our method in (22) and compare it to the popular conditional conformal prediction approach. We refer to App C.2 for a quick description of the principle underlying CCP. We trained an NCP model combined with an MLP architecture followed by whitening post-processing. See App C.2 for the full description. We obtained that

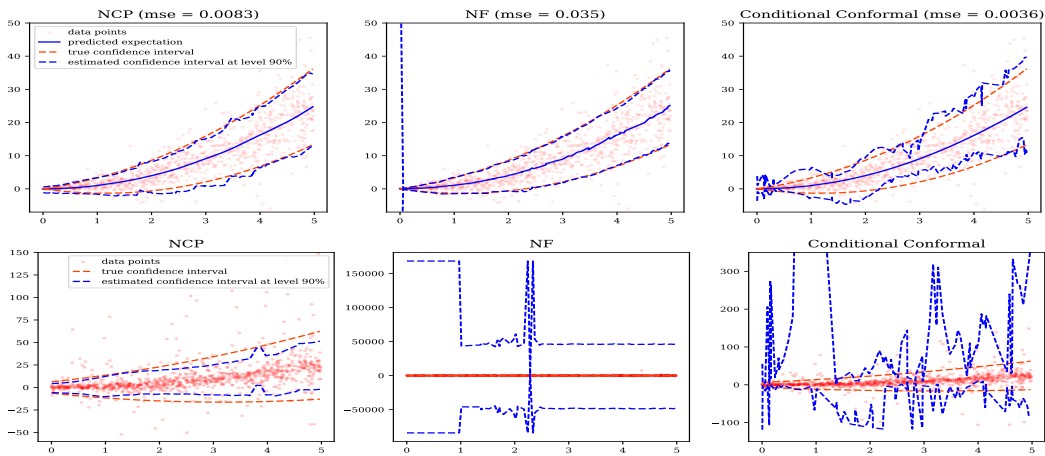

Figure 1: **Conditional mean (top only) and 90% confidence interval for NCP, NFs and CCP.** Top: Laplace distribution; Bottom: Cauchy distribution.

way the NCP conditional CDE model that we used according to (22) to build the conditional 90% confidence intervals. We proceeded similarly to build another set of conditional confidence intervals based on NFs. Finally, we also implemented the CCP method of Gibbs et al. (2023).

In Fig. 1, the marginal is $X \sim \mathrm{Unif}([0,5])$ and $Y|X = x$ follows either a Laplace distribution (top) with location and scale parameters $(\mu(x), b(x)) = (x^2, x)$ or a Cauchy distribution (bottom) with location and scale parameters $(x^2, 1 + x)$. In this experiment, we considered a favorable situation for the CCP method of Gibbs et al. (2023) by assuming prior knowledge that the true conditional location is a polynomial function (the truth is actually the square function). Every other parameter of the method was set as prescribed in their paper.

In Fig. 1, observe first that the CCP regression achieves the best estimation of the conditional mean $mse = 3.6 \cdot 10^{-3}$ against $mse = 3.8 \cdot 10^{-2}$ for NFs and $mse = 8.3 \cdot 10^{-3}$ for NCP, as expected since the CCP regression model is well-specified in this example. However, the CCP confidence intervals are unreliable for most of the considered conditioning. We also notice instability for NF and CCP when conditioning in the neighborhood of $x = 0$, with the NF confidence region exploding at $x = 0$. We suspect this is due to the fact that the conditional distribution at $x = 0$ is degenerate, hence violating the condition of existence of a diffeomorphism with the generating prior, a fundamental requirement for NFs models to work at all. Comparatively, NCP does not exhibit such instability around $x = 0$; it only tends to overestimate the confidence region for conditioning close to $x = 0$. The Cauchy distribution is known to be more challenging due to its heavy tail and undefined moments. In Fig 1 (bottom), we notice that NF and CCP completely collapse. This is not a surprising outcome since CCP relies on estimation of the mean which is undefined in this case, creating instability in the constructed confidence regions, while NF attempts to build a diffeomorphism between a Gaussian prior and the final Cauchy distribution. We suspect the conservative confidence region produced by NF might originate from the successive Jacobians involved in the NF mapping taking large values. In comparison, our NCP method still returns some reasonable results. Although the NCP coverage might appear underestimated for larger $x$, actual mean coverages computed on a test set of 200 samples are 88% for NCP, 99% for NF and 79% for CCP. Tab. 5 in Appendix C.2 provides a comparison study on real data for learning a confidence region with NCP, NF and a split conformal predictor featuring a Random Forest regressor (RFSCP).

**High-dimensional synthetic experiment** We simulated the following $d$-distribution for different values of $d \in \{100, 500, 1000\}$. Let $\bar{x} = (\bar{x}_1, \bar{x}_2, 0, \ldots, 0)^\top \in \mathbb{R}^d$ where $x' = (\bar{x}_1, \bar{x}_2)$ admits uniform distribution on the 2-dimensional unit sphere. We pick a random mapping $A \in \mathcal{O}_d$ and we set $X = A\bar{x}$ and the angle $\theta(X) = \arcsin(\bar{x}_2)$. Next we consider two conditional distribution models for $Y|X$ (Gaussian and discrete) described in Figure 3. NCP performs similarly to NF in the Gaussian case and outperforms NF for discrete distribution. Figure 7 in Appendix C.3 demonstrates that NCP scales effectively with increasing dimensionality $d$. As the dimension rises from $d = 100$

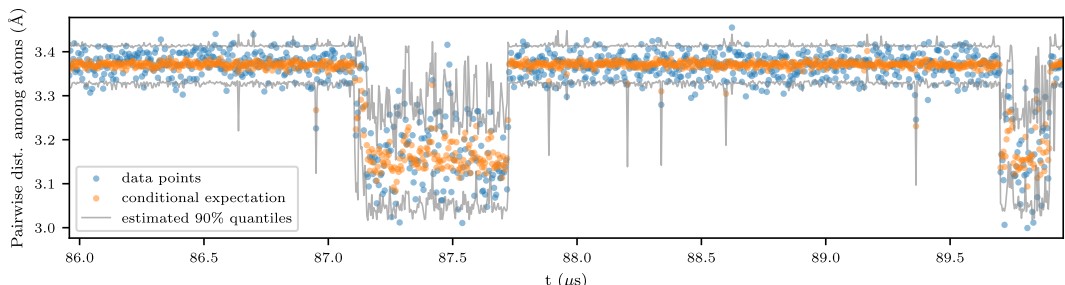

Figure 2: **Protein folding dynamics.** Pairwise Euclidean distances between Chignolin atoms exhibit increased variance during folded metastable states (between 87-88$\mu s$ and around 89.5$\mu s$). Ground truth is depicted in blue, predicted mean in orange, and the grey lines indicate the estimated 10% lower and upper quantiles.

to $d = 1000$, the computation time increases by only 20%, while maintaining strong statistical performance throughout.

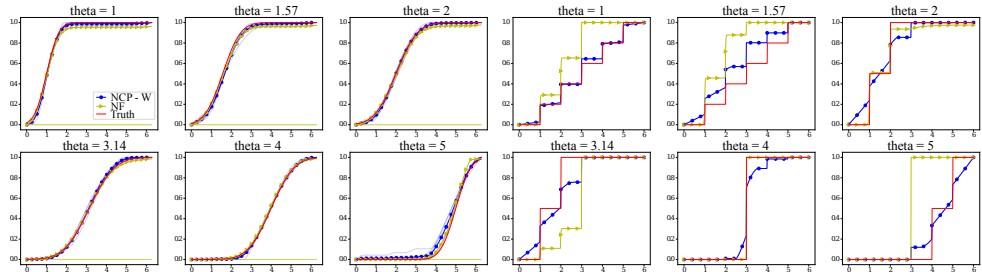

Figure 3: **High-dimensional synthetic experiment.** We consider two models for $Y|X$ with $d = 100$. **Left:** $Y|X \sim N(\theta(X), \sin(\theta(X)/2))$. **Right:** $Y \in \{1, 2, 3, 4, 5\}$ admits discrete distribution depending on $\theta(X)$: $Y|X \sim P_1$ if $\theta(X) \in [0, \pi/2)$, $P_2$ if $\theta(X) \in [\pi/2, \pi)$, $P_3$ if $\theta(X) \in [\pi, 3\pi/2)$, $P_4$ if $\theta(X) \in [3\pi/2, 2\pi)$. We take $P_1 = (1/5, 1/5, 1/5, 1/5, 1/5)$, $P_2 = (1/2, 1/2, 0, 0, 0)$, $P_3 = (0, 0, 1, 0, 0)$, $P_4 = (0, 0, 0, 1/2, 1/2)$.

**High-dimensional experiment in molecular dynamics** We investigate protein folding dynamics and predict conditional transition probabilities between metastable states. Figure 2 shows how, by integrating our NCP approach with a graph neural network (GNN), we achieve accurate state forecasting and strong uncertainty quantification, enabling efficient tracking of transitions. For further context and a full model description, see App C.3.

# 7 Conclusion

We introduced NCP, a novel neural operator approach to learn the conditional probability distribution from complex and highly nonlinear data. NCP offers a number of benefits. Notably, it streamlines the training process by requiring just one unconditional training phase to learn the joint distribution $p(x, y)$. Subsequently, it allows us to efficiently derive conditional probabilities and other relevant statistics from the trained model analytically, without any additional conditional training steps or Monte Carlo sampling. Additionally, our method is backed by theoretical non-asymptotic guarantees ensuring the soundness of our training method and the accuracy of the obtained conditional statistics. Our experiments on learning conditional densities and confidence regions demonstrate our approach's superiority or equivalence to leading methods, even using a simple Multi-Layer Perceptron (MLP) with two hidden layers and GELU activations. This highlights the effectiveness of a minimalistic architecture coupled with a theoretically grounded loss function. While complex architectures often dominate advanced machine learning, our results show that simplicity can achieve competitive results without compromising performance. Our numerical experiments suggest that, while our approach works well across different datasets and models, the price we pay for this generality appears to be the need for a relatively large sample size ($n \gtrsim 10^4$) to start outperforming other methods. Hence, a future direction is to study how to incorporate prior knowledge into our method to make it more data-efficient. Future works will also investigate the performance of NCP for multi-dimensional time series, causality and more general sensitivity analysis in uncertainty quantification.

## Acknowledgements

We acknowledge financial support from EU Project ELIAS under grant agreement No. 101120237, by NextGenerationEU and MUR PNRR project PE0000013 CUP J53C22003010006 "Future Artificial Intelligence Research (FAIR)" and by NextGenerationEU and MUR PNRR project RAISE "Robotics and AI for Socio-economic Empowerment" (ECS00000035).

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

# Supplemental material

The appendix is organized as follows:

- Appendix A provides additional details on the post-processing for NCP.

- Appendix B contains the proofs of the theoretical results and additional statistical results.

- In Appendix C, comprehensive details are presented regarding the experiment benchmark utilized to evaluate the performances of NCP.

## A  Details on training and algorithms

### A.1  Practical guidelines for training NCP

- It is better to choose a larger $d$ rather than a smaller one. Typically for the problems we considered in Section 6, we used $d \in \{100, 500\}$.

- The regularization parameter $\gamma$ was found to yield the best results for $\gamma \in \{10^{-2}, 10^{-3}\}$.

- To ensure the positivity of the singular values, we transform the vector $w^\theta$ with the Gaussian function $x \mapsto \exp(-x^2)$ to recover $\sigma^\theta$ during any call of the forward method. The vector $w^\theta$ is initialized at random with parameters following a normal distribution of mean 0 and standard deviation $1/d$.

- With the ReLU function, we observe instabilities in the loss function during training, whereas Tanh struggles to converge. In contrast, the use of GELU solves both problems.

- We can compute some statistical objects as a sanity check for the convergence of NCP training. For instance, we can ensure that the computed conditional CDFsatisfies all the conditions to be a valid CDF.

- After training, an additional post-processing may be applied to ensure the orthogonality of operators $u^\theta$ and $v^\theta$. This *whitening* step is described in Alg 2 in App A.3. It leads to an improvement of statistical accuracy of the trained NCP model. See the ablation study in Tab. 3.

### A.2  Learning dynamics with NCP

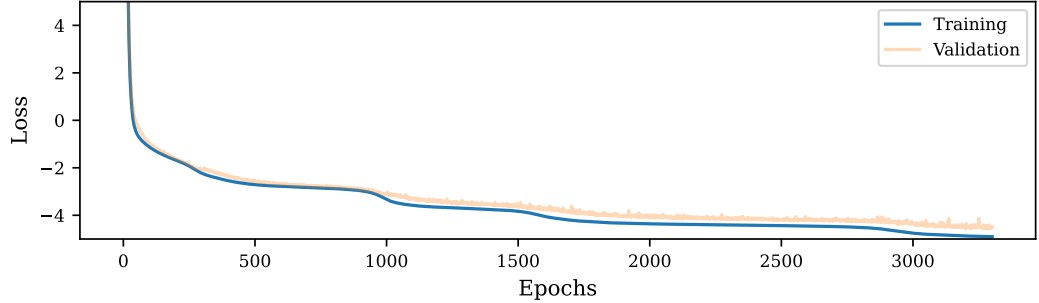

Figure 4: Learning dynamic for the Laplace experiment in Section 6.

### A.3  Whitening post-processing

We describe in Algorithm 2 the whitening post-processing procedure that we apply after training.

**Algorithm 2** Whitening procedure

---

**Require:** new data $(X_{\text{new}}, Y_{\text{new}})$; trained $u^\theta$, $\sigma^\theta$ and $v^\theta$
   Evaluate $u_X = u^\theta(X_{\text{train}})$ and $v_Y = v^\theta(Y_{\text{train}})$
   **Centering:**
      $u_X \leftarrow u_X - \hat{\mathsf{E}}(u^\theta(X_{\text{train}}))$ and $v_Y \leftarrow v_Y - \hat{\mathsf{E}}(v^\theta(Y_{\text{train}}))$
      $u_X \leftarrow u_X \text{diag}(\sigma^\theta)^{\frac{1}{2}}$ and $v_Y \leftarrow v_Y \text{diag}(\sigma^\theta)^{\frac{1}{2}}$
   Compute covariance matrices :
      $C_X \leftarrow u_x^\top u_x / n$
      $C_Y \leftarrow v_Y^\top v_Y / n$
      $C_{XY} \leftarrow u_X^\top v_Y / n$
   $U, V, \sigma^{\text{new}} \leftarrow \text{SVD}\left(C_X^{-1/2} C_{XY} C_Y^{-1/2}\right)$
   **if** $(X_{\text{new}}, Y_{\text{new}})$ is different than $(X_{\text{train}}, Y_{\text{train}})$ **then**
      $u_X \leftarrow \left(u^\theta(X_{\text{new}}) - \hat{\mathsf{E}}(u^\theta(X_{\text{train}}))\right)\text{diag}(\sigma^\theta)^{\frac{1}{2}}$
      $v_Y \leftarrow \left(v^\theta(X_{\text{new}}) - \hat{\mathsf{E}}(v^\theta(Y_{\text{train}}))\right)\text{diag}(\sigma^\theta)^{\frac{1}{2}}$
   **end if**
   Final whitening:
      $u_X^{\text{new}} \leftarrow u_X C_X^{-1/2} U$
      $v_Y^{\text{new}} \leftarrow v_Y C_Y^{-1/2} V$
   **return** $u_X^{\text{new}}$, $\sigma^{\text{new}}$, $v_Y^{\text{new}}$

---

# B   Proofs of theoretical results

## B.1   A reminder on Hilbert spaces and compact operators

**Definition 1.** *Given a vector space $\mathcal{H}$, we say it is a Hilbert space if there exists an inner product $\langle \cdot, \cdot \rangle$ such that:*

*$\mathcal{H}$ is complete with respect to the norm $\|x\| = \sqrt{\langle x, x \rangle}$ for all $x \in \mathcal{H}$.*

An important example of an infinite-dimensional Hilbert space is $L_\mu^2(\mathbb{R})$, the space of square-integrable functions w.r.t probability measure $\mu$ on $\mathbb{R}$ with the inner product defined as $\langle f, g \rangle = \int_{\mathbb{R}} f(x)\overline{g(x)}\, \mu(dx)$.

**Definition 2** (Bounded Operators)**.** *Let $\mathcal{H}_1$ and $\mathcal{H}_2$ be Hilbert spaces. A linear operator $T : \mathcal{H}_1 \to \mathcal{H}_2$ is called bounded if there exists a constant $C \geq 0$ such that for all $x \in \mathcal{H}_1$, the following inequality holds:*

$$\|Tx\|_{\mathcal{H}_2} \leq C\|x\|_{\mathcal{H}_1}.$$

*The smallest such constant $C$ is called the operator norm of $T$, denoted by $\|T\|$, and is given by:*

$$\|T\| = \sup_{x \neq 0} \frac{\|Tx\|_{\mathcal{H}_2}}{\|x\|_{\mathcal{H}_1}}.$$

Bounded operators are continuous and play a key role in functional analysis.

**Definition 3** (Compact Operators)**.** *Let $\mathcal{H}_1$ and $\mathcal{H}_2$ be Hilbert spaces. A bounded linear operator $T : \mathcal{H}_1 \to \mathcal{H}_2$ is called compact if for any bounded sequence $\{x_n\} \subset \mathcal{H}_1$, there exists a subsequence $\{x_{n_k}\}$ such that $Tx_{n_k}$ converges in $\mathcal{H}_2$.*

Compact operators can be viewed as infinite-dimensional analogues of matrices with finite rank in finite-dimensional spaces.

A key result in the theory of compact operators is the existence of a *singular value decomposition (SVD)* for compact operators. The following is the statement of the *Eckart-Young-Mirsky theorem*:

**Theorem 3** (Eckart-Young-Mirsky)**.** *Let $T : \mathcal{H}_1 \to \mathcal{H}_2$ be a compact operator between Hilbert spaces. Then $T$ can be decomposed as:*

$$T = \sum_{i=1}^{\infty} \sigma_i \langle \cdot, u_i \rangle v_i,$$

*where $\{u_i\} \subset \mathcal{H}_1$ and $\{v_i\} \subset \mathcal{H}_2$ are orthonormal sets, and $\sigma_i$ are the singular values of $T$, which satisfy $\sigma_1 \geq \sigma_2 \geq \cdots \geq 0$.*

*Moreover, for any rank-k operator $T_k = \sum_{i=1}^{k} \sigma_i \langle \cdot, u_i \rangle v_i$, we have:*

$$\|T - T_k\| = \min_{rank(S) \leq k} \|T - S\|,$$

*where $\|\cdot\|$ is the operator norm induced by the Hilbert spaces.*

## B.2 Proof of Lemma 1

*Proof of Lemma 1.* It follows from (3) and (5) that

$$\mathbb{P}[Y \in B \mid X \in A] - \mathbb{P}[Y \in B] - \frac{\langle \mathbb{1}_A, [\![\mathsf{D}_{Y|X}]\!]_d \mathbb{1}_B \rangle}{\mathbb{P}[X \in A]} = \frac{\langle \mathbb{1}_A, (\mathsf{D}_{Y|X} - [\![\mathsf{D}_{Y|X}]\!]_d) \mathbb{1}_B \rangle}{\mathbb{P}[X \in A]}.$$

Next, by definition of the operator norm, we have

$$|\langle \mathbb{1}_A, (\mathsf{D}_{Y|X} - [\![\mathsf{D}_{Y|X}]\!]_d) \mathbb{1}_B \rangle| \leq \|\mathsf{D}_{Y|X} - [\![\mathsf{D}_{Y|X}]\!]_d\|_{L^2_\nu(\mathcal{Y}) \to L^2_\mu(\mathcal{X})} \|\mathbb{1}_A\|_{L^2_\mu(\mathcal{X})} \|\mathbb{1}_B\|_{L^2_\nu(\mathcal{Y})}$$

$$= \|\mathsf{D}_{Y|X} - [\![\mathsf{D}_{Y|X}]\!]_d\|_{L^2_\nu(\mathcal{Y}) \to L^2_\mu(\mathcal{X})} \sqrt{\mathbb{P}[X \in A]} \sqrt{\mathbb{P}[Y \in B]},$$

where the operator norm $\|\mathsf{D}_{Y|X} - [\![\mathsf{D}_{Y|X}]\!]_d\|_{L^2_\nu(\mathcal{Y}) \to L^2_\mu(\mathcal{X})}$ is upper bounded by $\sigma^\star_{d+1}$ by definition of the SVD of $\mathsf{D}_{Y|X}$. $\qquad\square$

## B.3 Proof of Theorem 1

*Proof of Theorem 1.* In the following, to simplify notation, whenever dependency on the parameters is not crucial, recalling that $(X, Y)$ and $(X', Y')$ are two iid samples from the joint distribution $\rho$, we will denote the vector-valued random variables in the latent (embedding) space as $u := u^\theta(X)$, $u' := u^\theta(X')$, $v := v^\theta(Y)$ and $v' := v^\theta(Y')$, as well as $s = \sigma^\theta$ and $S := S_\theta$. Then, we can write the training loss simply as $\mathbb{E}\left[L_\gamma(u - \mathbb{E}u, u - \mathbb{E}u', v - \mathbb{E}v, v' - \mathbb{E}v', S)\right]$.

Let us further denote centered features as $\overline{u} = u - \mathbb{E}u$ and $\overline{v} = v - \mathbb{E}v$, and the operators based on them as $\overline{U}_\theta \colon \mathbb{R}^d \to L^2_\mu(\mathcal{X})$ and $\overline{V}_\theta \colon \mathbb{R}^d \to L^2_\nu(\mathcal{Y})$ by

$$\overline{U}_\theta z := z^\top (u^\theta - \mathbb{E}[u^\theta(X)]) \mathbb{1}_{\mathcal{X}} \quad \text{and} \quad \overline{V}_\theta z := z^\top (v^\theta - \mathbb{E}[v^\theta(Y)]) \mathbb{1}_{\mathcal{Y}}, \text{ for } z \in \mathbb{R}^d.$$

and prove that $\mathcal{L}_0(\theta) = \|\overline{U}_\theta S_\theta \overline{V}_\theta^*\|_{\mathrm{HS}}^2 - 2\,\mathrm{tr}(S_\theta \overline{U}_\theta^* \mathsf{D}_{Y|X} \overline{V}_\theta)$. Since we have that $\overline{U}_\theta = J_\mu U_\theta$ and $\overline{V}_\theta = J_\nu V_\theta$, where $J_\mu = I - \mathbb{1}_{\mathcal{X}} \otimes \mathbb{1}_{\mathcal{X}}$ and $J_\nu = I - \mathbb{1}_{\mathcal{Y}} \otimes \mathbb{1}_{\mathcal{Y}}$ are orthogonal projectors in $L^2_\mu(\mathcal{X})$ and $L^2_\nu(\mathcal{Y})$, respectively, as well as $\overline{U}_\theta^* \mathsf{D}_{Y|X} \overline{V}_\theta = U_\theta^* J_\mu \mathsf{D}_{Y|X} J_\nu V_\theta = U_\theta^* J_\mu \mathsf{E}_{Y|X} J_\nu V_\theta$, consequently

$$\overline{U}_\theta^* \mathsf{D}_{Y|X} \overline{V}_\theta = U_\theta^* \mathsf{E}_{Y|X} V_\theta - U_\theta^* \mathbb{1}_{\mathcal{X}} \otimes (V_\theta^* \mathbb{1}_{\mathcal{Y}}) = \mathbb{E}[u^\theta(X) \mathbb{E}[v^\theta(Y)^\top \mid X]] - (\mathbb{E}[u^\theta(X)])(\mathbb{E}[v^\theta(Y)])^\top,$$

that is $\overline{U}_\theta^* \mathsf{D}_{Y|X} \overline{V}_\theta = \mathbb{E}[uv^\top] - \mathbb{E}[u]\mathbb{E}[v]^\top$ is simply centered cross-covariance in the embedding space. Recalling that $U_\theta^* U_\theta = \mathbb{E}[uu^\top]$ and $V_\theta^* V_\theta = \mathbb{E}[vv^\top]$ are covariance matrices in the embedding space, similarly we get that $\overline{U}_\theta^* \overline{U}_\theta = \mathbb{E}[(u - \mathbb{E}u)(u - \mathbb{E}u)^\top]$ and $\overline{V}_\theta^* \overline{V}_\theta = \mathbb{E}[(v - \mathbb{E}v)(v - \mathbb{E}v)^\top]$ are centered covariances. Thus, we obtain

$$\mathcal{L}_0(\theta) = -2\,\mathrm{tr}\,\mathbb{E}[(S^{1/2}\overline{u})(S^{1/2}\overline{v})^\top] + \mathrm{tr}(\mathbb{E}[(S^{1/2}\overline{u})(S^{1/2}\overline{u})^\top] \mathbb{E}[(S^{1/2}\overline{v})(S^{1/2}\overline{v})^\top])$$

$$= -2\mathbb{E}[\overline{u}^\top S\overline{v}] + \mathrm{tr}(\mathbb{E}[(S^{1/2}\overline{u})(S^{1/2}\overline{u})^\top] \mathbb{E}[(S^{1/2}\overline{v})(S^{1/2}\overline{v})^\top])$$

which, by taking $(X, Y)$ and $(X', Y')$ to be iid random variables drawn from $\rho$, gives that $\mathcal{L}_0(\theta)$ can be written as

$$\mathbb{E}\left[-\overline{u}S\overline{v} - \overline{u}'S\overline{v}' + \overline{u}'S\overline{v} + \overline{u}S\overline{v}' + \tfrac{1}{2}\,\mathrm{tr}\left(S^{1/2}\overline{u}\overline{u}^\top S\overline{v}'\overline{v}'^\top S^{1/2} + S^{1/2}\overline{u}'\overline{u}'^\top S\overline{v}\overline{v}^\top S^{1/2}\right)\right]$$

$$= \mathbb{E}\left[\tfrac{1}{2}\left(\overline{u}^\top S\overline{v}'\right)^2 + \tfrac{1}{2}\left(\overline{u}'^\top S\overline{v}\right)^2 - (\overline{u} - \overline{u}')S(\overline{v} - \overline{v}')\right] = \mathbb{E}\left[L_0(\overline{u}, \overline{u}', \overline{v}, \overline{v}', s)\right]$$

$$= \mathbb{E}\left[L_0(u^\theta(X) - \mathbb{E}u^\theta(X), u^\theta(X') - \mathbb{E}u^\theta(X'), v^\theta(Y) - \mathbb{E}u^\theta(Y), v^\theta(Y') - \mathbb{E}u^\theta(Y'), \sigma^\theta)\right].$$

which implies that $\mathcal{L}_0(\theta) = \|\overline{U}_\theta S_\theta \overline{V}_\theta^*\|_{\mathrm{HS}}^2 - 2\operatorname{tr}(S_\theta \overline{U}_\theta^* \mathsf{D}_{Y|X} \overline{V}_\theta)$. Moreover, to show that $\mathcal{L}_\gamma(\theta) = \mathcal{L}_0(\theta) + \gamma\,\mathcal{R}(\theta)$. It suffices to note that

$$\|U_\theta^* U_\theta - I\|_F^2 = \operatorname{tr}((U_\theta^* U_\theta - I)^2) = \operatorname{tr}((U_\theta^* U_\theta)^2 - 2U_\theta^* U_\theta + I) =$$
$$= \operatorname{tr}(\mathbb{E}[uu^\top]\mathbb{E}[u'u'^\top] - \mathbb{E}[uu^\top] - \mathbb{E}[u'u'^\top] + I)$$
$$= \mathbb{E}[\operatorname{tr}(uu^\top u'u'^\top - uu^\top - u'u'^\top + I)] = \mathbb{E}\left[(u^\top u')^2 - \|u\|^2 - \|u'\|^2\right] + d,$$

as well as that $\|U_\theta^* \mathbb{1}_\mu\|^2 = \|\mathbb{E}u\|^2 = (\mathbb{E}u)^\top(\mathbb{E}u) = \mathbb{E}u^\top u'$, and apply the analogous reasoning for random variable $Y \sim \nu$.

Now, given $r > d+1$, let us denote $D_r := \sum_{i \in [r]} \sigma_i^\star u_i^\star \otimes v_i^\star$ and

$$\mathcal{L}_0^r(\theta) := \left\|D_r - \overline{U}_\theta S_\theta \overline{V}_\theta\right\|_{\mathrm{HS}}^2 - \|D_r\|_{\mathrm{HS}}^2. \tag{23}$$

Then, applying the Eckhart-Young-Mirsky theorem, we obtain that

$$\mathcal{L}_0^r(\theta) \geq \sum_{i=d+1}^r \sigma_i^{\star 2} - \sum_{i \in [r]} \sigma_i^{\star 2} = -\sum_{i \in [d]} \sigma_i^{\star 2},$$

with equality holding whenever $(\sigma_i^\theta, u_i^\theta, v_i^\theta) = (\sigma_i^\star, u_i^\star, v_i^\star)$, $\rho$-almost everywhere.

To prove that the same holds for $\mathcal{L}_0(\theta)$, observe that after expanding the HS norm via trace in (23), we have that

$$\mathcal{L}_0^r(\theta) = -2\operatorname{tr}\left(S_\theta^{1/2}\overline{U}_\theta^* D_r \overline{V}_\theta S_\theta^{1/2}\right) + \|U_\theta S_\theta V_\theta^*\|_{\mathrm{HS}}^2,$$

and, consequently,

$$\mathcal{L}_0^r(\theta) = \|\overline{U}_\theta S_\theta \overline{V}_\theta^*\|_{\mathrm{HS}}^2 - 2\operatorname{tr}(S_\theta^{1/2}\overline{U}_\theta^* D_r \overline{V}_\theta S_\theta^{1/2}) = \mathcal{L}_0(\theta) + 2\operatorname{tr}(S_\theta \overline{U}_\theta^* (\mathsf{D}_{Y|X} - D_r)\overline{V}_\theta).$$

Thus, using Cauchy-Schwartz inequality, we obtain

$$|\mathcal{L}_0^r(\theta) - \mathcal{L}_0(\theta)| \leq |\operatorname{tr}(S_\theta \overline{U}_\theta^* (\mathsf{D}_{Y|X} - D_r)\overline{V}_\theta)| \leq \|S_\theta\| \|\overline{U}_\theta^*\|_{\mathrm{HS}} \|\mathsf{D}_{Y|X} - [\![\mathsf{D}_{Y|X}]\!]_r \| \|\overline{V}_\theta^*\|_{\mathrm{HS}},$$

and, therefore, $|\mathcal{L}_0^r(\theta) - \mathcal{L}_0(\theta)| \leq \sigma_{r+1}^\star \sqrt{\operatorname{tr}(U_\theta^* U_\theta)\operatorname{tr}(V_\theta^* V_\theta)} \leq Md\sigma_{r+1}^\star$, where the constant is given by $M := \max_{i \in [d]}\{\|u_i^\theta - \mathbb{E}u_i^\theta\|_{L_\mu^2(\mathcal{X})}, \|v_i^\theta - \mathbb{E}v_i^\theta\|_{L_\nu^2(\mathcal{Y})}\} < \infty$. So, $\mathcal{L}_0^r(\theta) - Md\sigma_{r+1}^\star \leq \mathcal{L}_0(\theta) \leq \mathcal{L}_0^r(\theta) + Md\sigma_{r+1}^\star$, and, since $r > d+1$ was arbitrary, we can take $r$ arbitrary large to obtain $\sigma_r^\star \to 0$ and conclude that $\mathcal{L}_0(\theta) \geq -\sum_{i \in [d]} \sigma_i^{\star 2}$, with equality holding when $(\sigma_i^\theta, u_i^\theta, v_i^\theta) = (\sigma_i^\star, u_i^\star, v_i^\star)$, $\rho$-almost everywhere, since then $\overline{U}_\theta^* \mathsf{D}_{Y|X}\overline{V}_\theta = U_\theta^* \mathsf{D}_{Y|X} V_\theta = U_\theta^* U_\theta S_\theta = S_\theta = \operatorname{diag}(\sigma_1, \ldots, \sigma_d)$.

Finally, we prove that $\gamma > 0$ and $\sigma_d^\star > \sigma_{d+1}^\star$ assure uniqueness of the global optimum. First, if the global minimum is achieved $\sigma_d^\star > \sigma_{d+1}^\star$ allows one to use uniqueness result in the Eckhart-Young-Mirsky theorem that states that $\sum_{i \in [d]} \sigma_i^\star u_i^\star \otimes \widehat{v}_i^\theta = \sum_{i \in [d]} \sigma_i^\theta \widehat{u}_i^\theta \otimes \widehat{v}_i^\theta$. But since, $\gamma > 0$ implies that $\mathcal{R}(\theta) = 0$, i.e. $(u_i^\theta)_{i \in [d]} \subset L_\mu^2(\mathcal{X})$ and $(v_i^\theta)_{i \in [d]} \subset L_\nu^2(\mathcal{Y})$ are two orthonormal systems in the corresponding orthogonal complements of constant functions, using the uniqueness of SVD, the proof is completed. $\qquad\square$

## B.4 Proof of Theorem 2

*Proof of Theorem 2.* Let us denote the operators arising from centered and empirically centered features as $\overline{U}_\theta, \widehat{U}_\theta \colon \mathbb{R}^d \to L_\mu^2(\mathcal{X})$ and $\widehat{V}_\theta, \overline{V}_\theta \colon \mathbb{R}^d \to L_\nu^2(\mathcal{Y})$ by

$$\overline{U}_\theta z := z^\top(u^\theta - \mathbb{E}[u^\theta(X)])\mathbb{1}_\mathcal{X}, \quad \overline{V}_\theta z := z^\top(v^\theta - \mathbb{E}[v^\theta(Y)])\mathbb{1}_\mathcal{Y} \quad \text{and} \quad \widehat{U}_\theta z := z^\top \widehat{u}^\theta, \quad \widehat{V}_\theta z := z^\top \widehat{v}^\theta,$$

respectively, for $z \in \mathbb{R}^d$.

We first bound the error of the conditional expectation model as $\|\mathsf{D}_{Y|X} - \widehat{\mathsf{D}}_{Y|X}^\theta\|$ as follows.

$$\|\mathsf{D}_{Y|X} - \widehat{\mathsf{D}}_{Y|X}^\theta\| = \|\mathsf{D}_{Y|X} \pm [\![\mathsf{D}_{Y|X}]\!]_d \pm U_\theta^* S_\theta V_\theta \pm \overline{U}_\theta^* S_\theta \overline{V}_\theta - \widehat{U}_\theta^* S_\theta \widehat{V}_\theta\|$$
$$\leq \sigma_{d+1}^\star + \mathcal{E}_\theta + \|U_\theta^* S_\theta V_\theta - \overline{U}_\theta^* S_\theta \overline{V}_\theta\| + \|\overline{U}_\theta^* S_\theta \overline{V}_\theta - \widehat{U}_\theta^* S_\theta \widehat{V}_\theta\|.$$

Next, using that $\|S_\theta\| \le 1$ and that centered covariances are bounded by uncentered ones, i.e. $\overline{U}_\theta^* \overline{U}_\theta \preceq U_\theta^* U_\theta$, we have

$$\begin{aligned}
\|U_\theta^* S_\theta V_\theta - \overline{U}_\theta^* S_\theta \overline{V}_\theta\| &= \|U_\theta^* S_\theta V_\theta \pm \overline{U}_\theta^* S_\theta V_\theta - \overline{U}_\theta^* S_\theta \overline{V}_\theta\| \\
&\le \|U_\theta - \overline{U}_\theta\|\|V_\theta\| + \|\overline{U}_\theta\|\|V_\theta - \overline{V}_\theta\| \\
&\le \|U_\theta^* \mathbb{1}_{\mathcal{X}}\|\|V_\theta^* V_\theta\|^{1/2} + \|V_\theta^* \mathbb{1}_{\mathcal{Y}}\|\|U_\theta^* U_\theta\|^{1/2} \le 2\mathcal{E}_\theta \sqrt{1 + \mathcal{E}_\theta}.
\end{aligned}$$

In a similar way, we obtain

$$\begin{aligned}
\|\overline{U}_\theta^* S_\theta \overline{V}_\theta - \widehat{U}_\theta^* S_\theta \widehat{V}_\theta\| &= \|\overline{U}_\theta^* S_\theta \overline{V}_\theta \pm \widehat{U}_\theta^* S_\theta \overline{V}_\theta - \widehat{U}_\theta^* S_\theta \widehat{V}_\theta\| \\
&\le \|\overline{U}_\theta - \widehat{U}_\theta\|\|\overline{V}_\theta\| + \|\widehat{U}_\theta\|\|\overline{V}_\theta - \widehat{V}_\theta\| \\
&\le \|\overline{U}_\theta - \widehat{U}_\theta\|\|\overline{V}_\theta\| + \|\overline{V}_\theta - \widehat{V}_\theta\|\|\overline{U}_\theta\| + \|\overline{U}_\theta - \widehat{U}_\theta\|\|\overline{V}_\theta - \widehat{V}_\theta\| \\
&\le \sqrt{1 + \mathcal{E}_\theta}\big(\|\widehat{\mathsf{E}}_x[u^\theta] - \mathbb{E}[u^\theta(X)]\| + \|\widehat{\mathsf{E}}_y[v^\theta] - \mathbb{E}[v^\theta(Y)]\|\big) \\
&\qquad + \|\widehat{\mathsf{E}}_x[u^\theta] - \mathbb{E}[u^\theta(X)]\|\,\|\widehat{\mathsf{E}}_y[v^\theta] - \mathbb{E}[v^\theta(Y)]\| \\
&\le 2\sqrt{1 + \mathcal{E}_\theta}\,\varepsilon_n(\delta) + [\varepsilon_n(\delta)]^2.
\end{aligned}$$

where $\|\widehat{\mathsf{E}}_x[u^\theta] - \mathbb{E}[u^\theta(X)]\| \le \varepsilon_n(\delta)$ and $\|\widehat{\mathsf{E}}_y[v^\theta] - \mathbb{E}[v^\theta(Y)]\| \le \varepsilon_n(\delta)$ hold w.p.a.l. $1 - \delta$ in view of Lemma 2.

To summarize, it holds w.p.a.l. $1 - \delta$

$$\|\mathsf{D}_{Y|X} - \widehat{\mathsf{D}}_{Y|X}^\theta\| \le \sigma_{d+1}^\star + \mathcal{E}_\theta + 2\sqrt{1 + \mathcal{E}_\theta}(\mathcal{E}_\theta + \varepsilon_n(\delta)) + [\varepsilon_n(\delta)]^2 =: \psi_n(\delta). \tag{24}$$

By definition in (5) and (14), we have

$$\mathbb{P}[Y \in B \mid X \in A] - \widehat{p}_\theta(B \mid A) = \mathbb{E}[\mathbb{1}_B(Y)] - \widehat{\mathsf{E}}_y[\mathbb{1}_B] + \frac{\langle \mathbb{1}_A, \mathsf{D}_{Y|X}\mathbb{1}_B\rangle}{\mathbb{E}[\mathbb{1}_A(X)]} - \frac{\langle \mathbb{1}_A, \widehat{\mathsf{D}}_{Y|X}^\theta \mathbb{1}_B\rangle}{\widehat{\mathsf{E}}_x[\mathbb{1}_A]},$$

and

$$\frac{\langle \mathbb{1}_A, \mathsf{D}_{Y|X}\mathbb{1}_B\rangle}{\mathbb{E}[\mathbb{1}_A(X)]} = \frac{\langle \mathbb{1}_A, (\mathsf{D}_{Y|X} - \widehat{\mathsf{D}}_{Y|X}^\theta)\mathbb{1}_B\rangle}{\mathbb{E}[\mathbb{1}_A(X)]} + \frac{\langle \mathbb{1}_A, \widehat{\mathsf{D}}_{Y|X}^\theta \mathbb{1}_B\rangle}{\widehat{\mathsf{E}}_x[\mathbb{1}_A]} \frac{\widehat{\mathsf{E}}_x[\mathbb{1}_A]}{\mathbb{E}[\mathbb{1}_A(X)]}.$$

Note also that $\|\mathbb{1}_A(X)\|_{L_\mu^2(\mathcal{X})} = \sqrt{\mathbb{E}[\mathbb{1}_A(X)]} = \sqrt{\mathbb{P}[X \in A]}$, $\|\mathbb{1}_B(Y)\|_{L_\nu^2(\mathcal{Y})} = \sqrt{\mathbb{E}[\mathbb{1}_B(Y)]} = \sqrt{\mathbb{P}[Y \in B]}$, for any $A \in \Sigma_{\mathcal{X}}$ and $B \in \Sigma_{\mathcal{Y}}$ and

$$|\langle \mathbb{1}_A, (\mathsf{D}_{Y|X} - \widehat{\mathsf{D}}_{Y|X}^\theta)\mathbb{1}_B\rangle| \le \|\mathsf{D}_{Y|X} - \widehat{\mathsf{D}}_{Y|X}^\theta\|\|\mathbb{1}_A(X)\|_{L_\mu^2(\mathcal{X})}\|\mathbb{1}_B(Y)\|_{L_\nu^2(\mathcal{Y})}.$$

Combining the previous observations, we get

$$\begin{aligned}
|\mathbb{P}[Y \in B \mid X \in A] - \widehat{p}_\theta(B \mid A)| &\le \left(\frac{|\widehat{\mathsf{E}}_y[\mathbb{1}_B] - \mathbb{E}[\mathbb{1}_B(Y)]|}{\mathbb{E}[\mathbb{1}_B(Y)]} + \frac{\|\mathsf{D}_{Y|X} - \widehat{\mathsf{D}}_{Y|X}^\theta\|}{\sqrt{\mathbb{E}[\mathbb{1}_A(X)]\mathbb{E}[\mathbb{1}_B(Y)]}}\right)\mathbb{E}[\mathbb{1}_B(Y)] \\
&\qquad + \frac{|\langle \mathbb{1}_A, \widehat{\mathsf{D}}_{Y|X}^\theta \mathbb{1}_B\rangle|}{\widehat{\mathsf{E}}_x[\mathbb{1}_A]} \frac{|\widehat{\mathsf{E}}_x[\mathbb{1}_A] - \mathbb{E}[\mathbb{1}_A(X)]|}{\mathbb{E}[\mathbb{1}_A(X)]}, \tag{25}
\end{aligned}$$

and

$$\begin{aligned}
\frac{|\langle \mathbb{1}_A, \widehat{\mathsf{D}}_{Y|X}^\theta \mathbb{1}_B\rangle|}{\widehat{\mathsf{E}}_x[\mathbb{1}_A]} &\le \frac{\mathbb{E}[\mathbb{1}_A(X)]}{\widehat{\mathsf{E}}_x[\mathbb{1}_A]}\left(\frac{|\langle \mathbb{1}_A, \mathsf{D}_{Y|X}\mathbb{1}_B\rangle|}{\mathbb{E}[\mathbb{1}_A(X)]} + \frac{|\langle \mathbb{1}_A, (\mathsf{D}_{Y|X} - \widehat{\mathsf{D}}_{Y|X}^\theta)\mathbb{1}_B\rangle|}{\mathbb{E}[\mathbb{1}_A(X)]}\right) \\
&\le \frac{\mathbb{E}[\mathbb{1}_A(X)]}{\widehat{\mathsf{E}}_x[\mathbb{1}_A]}\left(\|\mathsf{D}_{Y|X}\| + \|\mathsf{D}_{Y|X} - \widehat{\mathsf{D}}_{Y|X}^\theta\|\right)\sqrt{\frac{\mathbb{E}[\mathbb{1}_B(Y)]}{\mathbb{E}[\mathbb{1}_A(X)]}} \\
&\le \frac{\mathbb{E}[\mathbb{1}_A(X)]}{\widehat{\mathsf{E}}_x[\mathbb{1}_A]}\sqrt{\frac{\mathbb{E}[\mathbb{1}_B(Y)]}{\mathbb{E}[\mathbb{1}_A(X)]}}\left(1 + \|\mathsf{D}_{Y|X} - \widehat{\mathsf{D}}_{Y|X}^\theta\|\right), \tag{26}
\end{aligned}$$

where we have used that $\|\mathsf{D}_{Y|X}\| \leq 1$.

Similarly, we have

$$
\frac{\mathbb{P}[Y \in B \mid X \in A]}{\mathbb{P}[Y \in B]} - \frac{\widehat{p}_\theta(B \mid A)}{\widehat{p}_y(B)}
$$

$$
= \frac{\langle \mathbb{1}_A, \mathsf{D}_{Y|X}\mathbb{1}_B \rangle}{\mathbb{E}[\mathbb{1}_A(X)]\mathbb{E}[\mathbb{1}_B(Y)]} - \frac{\langle \mathbb{1}_A, \widehat{\mathsf{D}}^\theta_{Y|X}\mathbb{1}_B \rangle}{\widehat{\mathsf{E}}_x[\mathbb{1}_A]\widehat{\mathsf{E}}_y[\mathbb{1}_B]}
$$

$$
= \frac{\langle \mathbb{1}_A, (\mathsf{D}_{Y|X} - \widehat{\mathsf{D}}^\theta_{Y|X})\mathbb{1}_B \rangle}{\widehat{\mathsf{E}}_x[\mathbb{1}_A]\widehat{\mathsf{E}}_y[\mathbb{1}_B]}
$$

$$
+ \langle \mathbb{1}_A, \widehat{\mathsf{D}}^\theta_{Y|X}\mathbb{1}_B \rangle \left( \frac{1}{\mathbb{E}[\mathbb{1}_A(X)]\mathbb{E}[\mathbb{1}_B(Y)]} - \frac{1}{\widehat{\mathsf{E}}_x[\mathbb{1}_A]\widehat{\mathsf{E}}_y[\mathbb{1}_B]} \right)
$$

$$
= \frac{\langle \mathbb{1}_A, (\mathsf{D}_{Y|X} - \widehat{\mathsf{D}}^\theta_{Y|X})\mathbb{1}_B \rangle}{\widehat{\mathsf{E}}_x[\mathbb{1}_A]\widehat{\mathsf{E}}_y[\mathbb{1}_B]}
$$

$$
+ \frac{\langle \mathbb{1}_A, \widehat{\mathsf{D}}^\theta_{Y|X}\mathbb{1}_B \rangle}{\mathbb{E}[\mathbb{1}_A(X)]\mathbb{E}[\mathbb{1}_B(Y)]} \left( \frac{(\widehat{\mathsf{E}}_x[\mathbb{1}_A] - \mathbb{E}[\mathbb{1}_A(X)])\widehat{\mathsf{E}}_y[\mathbb{1}_B] + \mathbb{E}[\mathbb{1}_A(X)](\widehat{\mathsf{E}}_y[\mathbb{1}_B] - \mathbb{E}[\mathbb{1}_B(Y)])}{\widehat{\mathsf{E}}_x[\mathbb{1}_A]\widehat{\mathsf{E}}_y[\mathbb{1}_B]} \right)
$$

$$
= \frac{\langle \mathbb{1}_A, (\mathsf{D}_{Y|X} - \widehat{\mathsf{D}}^\theta_{Y|X})\mathbb{1}_B \rangle}{\widehat{\mathsf{E}}_x[\mathbb{1}_A]\widehat{\mathsf{E}}_y[\mathbb{1}_B]}
$$

$$
+ \frac{\langle \mathbb{1}_A, \widehat{\mathsf{D}}^\theta_{Y|X}\mathbb{1}_B \rangle}{\mathbb{E}[\mathbb{1}_A(X)]\mathbb{E}[\mathbb{1}_B(Y)]} \left( \frac{\widehat{\mathsf{E}}_x[\mathbb{1}_A] - \mathbb{E}[\mathbb{1}_A(X)]}{\widehat{\mathsf{E}}_x[\mathbb{1}_A]} + \frac{\mathbb{E}[\mathbb{1}_A(X)](\widehat{\mathsf{E}}_y[\mathbb{1}_B] - \mathbb{E}[\mathbb{1}_B(Y)])}{\widehat{\mathsf{E}}_x[\mathbb{1}_A]\widehat{\mathsf{E}}_y[\mathbb{1}_B]} \right). \quad (27)
$$

Next Lemmas 3 and 4 combined with (18) and elementary algebra give w.p.a.l. $1 - 2\delta$ that

$$
\left| \frac{\widehat{\mathsf{E}}_x[\mathbb{1}_A] - \mathbb{E}[\mathbb{1}_A(X)]}{\widehat{\mathsf{E}}_x[\mathbb{1}_A]} \right| \leq 2\varphi_X(A)\bar{\epsilon}_n(\delta), \quad \left| \frac{\widehat{\mathsf{E}}_y[\mathbb{1}_B] - \mathbb{E}[\mathbb{1}_B(Y)]}{\widehat{\mathsf{E}}_x[\mathbb{1}_B]} \right| \leq 2\varphi_Y(B)\bar{\epsilon}_n(\delta),
$$

and

$$
\frac{\mathbb{E}[\mathbb{1}_A(X)]}{\widehat{\mathsf{E}}_x[\mathbb{1}_A]} \vee \frac{\mathbb{E}[\mathbb{1}_B(Y)]}{\widehat{\mathsf{E}}_y[\mathbb{1}_B]} \leq 2, \quad \left| \frac{\mathbb{E}[\mathbb{1}_A(X)](\widehat{\mathsf{E}}_y[\mathbb{1}_B] - \mathbb{E}[\mathbb{1}_B(Y)])}{\widehat{\mathsf{E}}_x[\mathbb{1}_A]\widehat{\mathsf{E}}_y[\mathbb{1}_B]} \right| \leq 4\varphi_Y(B)\bar{\epsilon}_n(\delta).
$$

It also holds on the same probability event as above that

$$
\left| \frac{\langle \mathbb{1}_A, (\mathsf{D}_{Y|X} - \widehat{\mathsf{D}}^\theta_{Y|X})\mathbb{1}_B \rangle}{\widehat{\mathsf{E}}_x[\mathbb{1}_A]\widehat{\mathsf{E}}_y[\mathbb{1}_B]} \right| \leq \frac{\|\mathsf{D}_{Y|X} - \widehat{\mathsf{D}}^\theta_{Y|X}\|}{\sqrt{\mathbb{E}[\mathbb{1}_A(X)]\mathbb{E}[\mathbb{1}_B(Y)]}} \frac{\mathbb{E}[\mathbb{1}_A(X)]}{\widehat{\mathsf{E}}_x[\mathbb{1}_A]} \frac{\mathbb{E}[\mathbb{1}_B(Y)]}{\widehat{\mathsf{E}}_y[\mathbb{1}_B]} \leq 4 \frac{\|\mathsf{D}_{Y|X} - \widehat{\mathsf{D}}^\theta_{Y|X}\|}{\sqrt{\mathbb{E}[\mathbb{1}_A(X)]\mathbb{E}[\mathbb{1}_B(Y)]}}.
$$

Combining Lemma 2 and (24), we get with probability at least $1 - \delta$ that $\|\mathsf{D}_{Y|X} - \widehat{\mathsf{D}}^\theta_{Y|X}\| \leq \psi_n(\delta)$.

By a union bound combining the last two displays with (24), (27), (25) and (26), we get with probability at least $1 - 3\delta$

$$
\left| \frac{\mathbb{P}[Y \in B \mid X \in A]}{\mathbb{P}[Y \in B]} - \frac{\widehat{p}_\theta(B \mid A)}{\widehat{p}_y(B)} \right| \leq \frac{4\psi_n(\delta) + [1 + \psi_n(\delta)] [2\varphi_X(A) + 4\varphi_Y(B)] \bar{\epsilon}_n(\delta)}{\sqrt{\mathbb{E}[\mathbb{1}_A(X)]\mathbb{E}[\mathbb{1}_B(Y)]}}, \quad (28)
$$

and

$$
\left| \frac{\mathbb{P}[Y \in B \mid X \in A] - \widehat{p}_\theta(B \mid A)}{\mathbb{P}[Y \in B]} \right| \leq \varphi_Y(B)\bar{\epsilon}_n(\delta) + \frac{2(1 + \psi_n(\delta))\varphi_X(A)\bar{\epsilon}_n(\delta) + \psi_n(\delta)}{\sqrt{\mathbb{E}[\mathbb{1}_A(X)]\mathbb{E}[\mathbb{1}_B(Y)]}}. \quad (29)
$$

Replacing $\delta$ by $\delta/3$, we get the result w.p.a.l. $1 - \delta$.

$\square$

The following result will be useful to investigate the theoretical properties of the NCP method in the iid setting.

**Lemma 2.** *Let Assumption 1 be satisfied. Assume in addition that $n \geq c_u^2 d$. Then there exists an absolute constant $C > 0$ such that, for any $\delta \in (0,1)$, it holds w.p.a.l. $1 - \delta$*

$$\|\widehat{\mathsf{E}}_x[u^\theta] - \mathbb{E}[u^\theta(X)]\| \leq C\, c_u \sqrt{d}\left(\frac{\log(e\delta^{-1})}{n} + \sqrt{\frac{\log(e\delta^{-1})}{n}}\right).$$

*Similarly, if $n \geq c_v^2 d$, w.p.a.l. $1 - \delta$*

$$\|\widehat{\mathsf{E}}_y[v^\theta] - \mathbb{E}[v^\theta(Y)]\| \leq C c_v \sqrt{d}\left(\frac{\log(e\delta^{-1})}{n} + \sqrt{\frac{\log(e\delta^{-1})}{n}}\right).$$

*Proof of Lemma 2.* We note that

$$\widehat{\mathsf{E}}_x[u^\theta] - \mathbb{E}[u^\theta(X)] = \frac{1}{n}\sum_{i=1}^n Z_i \quad \text{with} \quad Z_i = u^\theta(X_i) - \mathbb{E}u^\theta(X_i), \ \forall i \in [n].$$

We note that $\|Z_i\| \leq 2c_u \sqrt{d} =: U$ and $\mathrm{Var}(Z_i) = \mathrm{Var}(\|u^\theta(X_i) - \mathbb{E}[u^\theta(X_i)]\|) = \sigma_\theta^2(X)$ for any $i \in [n]$. We apply Minsker (2017, Corollary 4.1) to get for any $t \geq \frac{1}{6}(U + \sqrt{U^2 + 36n\sigma_\theta^2(X)})$,

$$\mathbb{P}\left[\|\sum_{i=1}^n Z_i\| > t\right] \leq 28\exp\left(-\frac{t^2/2}{n\sigma_\theta^2(X) + tU/3}\right). \tag{30}$$

Replacing $t$ by $nt$ and some elementary algebra give for any $t \geq \frac{1}{6}\left(\frac{U}{n} + \sqrt{\frac{U^2}{n^2} + 36\frac{\sigma_\theta^2(X)}{n}}\right) =: \overline{c}$, w.p.a.l. $1 - 28\exp(-t)$,

$$\|\frac{1}{n}\sum_{i=1}^n Z_i\| \leq \frac{4U}{3}\frac{t}{n} + 2\sigma_\theta(X)\sqrt{\frac{t}{n}}.$$

Replacing $t$ by $t + \overline{c}$, we get for any $t \geq 0$, w.p.a.l. $1 - 28\exp(-t + \overline{c})$,

$$\|\frac{1}{n}\sum_{i=1}^n Z_i\| \leq \frac{4U}{3}\frac{t + \overline{c}}{n} + 2\sigma_\theta(X)\sqrt{\frac{t + \overline{c}}{n}}.$$

Up to a rescaling of the constants, there exists a numerical constant $C > 0$ such that for any $\delta \in (0,1)$, w.p.a.l. $1 - \delta$

$$\|\frac{1}{n}\sum_{i=1}^n Z_i\| \leq C\left(\frac{U}{n}\overline{c} + \sigma_\theta(X)\sqrt{\frac{\overline{c}}{n}} + U\frac{t}{n} + \sigma_\theta(X)\sqrt{\frac{t}{n}}\right).$$

Elementary computations give the following bound, that is, there exists a numerical constant $C > 0$ such that for any $t > 0$, w.p.a.l. $1 - \exp(-t)$

$$\|\frac{1}{n}\sum_{i=1}^n Z_i\| \leq C\left(\frac{c_u\sqrt{d}}{n} \vee \frac{c_u^2 d}{n^2} \vee \frac{\sigma_\theta^{3/2}(X)}{n^{3/4}} \vee \frac{\sigma_\theta^2(X)}{n} + c_u\frac{\sqrt{d}t}{n} + \sigma_\theta(X)\sqrt{\frac{t}{n}}\right).$$

Under Assumption 1 and the condition $\frac{c_u^2 d}{n} \leq 1$, it also holds that $\frac{\sigma_\theta^2(X)}{n} \leq 1$ since $\sigma_\theta^2(X) \leq c_u^2 d$. Consequently, the bound simplifies and we obtain for any $t > 1$, w.p.a.l. $1 - \exp(-t)$

$$\|\frac{1}{n}\sum_{i=1}^n Z_i\| \leq C\, c_u \sqrt{d}\left(\frac{t}{n} \vee \sqrt{\frac{t}{n}}\right),$$

where $C > 0$ is possibly a different absolute constant from the previous bound. Taking $t = \log e\delta^{-1}$ for any $\delta \in (0,1)$ gives the first result. We proceed similarly to get the second result. □

**Control on empirical probabilities**    We derive now a concentration result for empirical probabilities.

**Lemma 3.** *For any $A \in \Sigma_{\mathcal{X}}$ and any $\delta \in (0,1)$, it holds w.p.a.l. $1 - \delta$*

$$|\widehat{\mathsf{E}}_x[\mathbb{1}_A] - \mathbb{E}[\mathbb{1}_A(X)]| \leq 2\frac{\log 2\delta^{-1}}{n} + \sqrt{\mathbb{P}[X \in A](1 - \mathbb{P}[X \in A])}\sqrt{2\frac{\log 2\delta^{-1}}{n}}.$$

*Assume in addition that $\mathbb{P}(X \in A) \geq 2\sqrt{2\frac{\log 2\delta^{-1}}{n}}$. Then it holds w.p.a.l. $1 - \delta$*

$$\frac{|\widehat{\mathsf{E}}_x[\mathbb{1}_A] - \mathbb{E}[\mathbb{1}_A(X)]|}{\mathbb{E}[\mathbb{1}_A(X)]} \leq \sqrt{2\frac{\log 2\delta^{-1}}{n}}\sqrt{1 \vee \frac{1 - \mathbb{P}[X \in A]}{\mathbb{P}[X \in A]}}.$$

*Proof.* We note that

$$\widehat{\mathsf{E}}_x[\mathbb{1}_A(X)] - \mathbb{E}[\mathbb{1}_A(X)] = \frac{1}{n}\sum_{i=1}^{n} Z_i \quad \text{with} \quad Z_i = \mathbb{1}_A(X_i) - \mathbb{E}[\mathbb{1}_A(X_i)], \ \forall i \in [n].$$

We note that $|Z_i| \leq 2$ and $\mathrm{Var}(Z_i) = \mathbb{P}[X \in A](1 - \mathbb{P}[X \in A])$. Then Bercu et al. (2015, Theorem 2.9) gives w.p.a.l. $1 - 2\delta$

$$|\widehat{\mathsf{E}}_x[\mathbb{1}_A] - \mathbb{E}[\mathbb{1}_A(X)]| \leq 2\frac{\log \delta^{-1}}{n} + \sqrt{\mathbb{P}[X \in A](1 - \mathbb{P}[X \in A])}\sqrt{2\frac{\log \delta^{-1}}{n}}.$$

Dividing by $\mathbb{E}[\mathbb{1}_A(X)]$ gives w.p.a.l. $1 - 2\delta$

$$\frac{|\widehat{\mathsf{E}}_x[\mathbb{1}_A] - \mathbb{E}[\mathbb{1}_A(X)]|}{\mathbb{E}[\mathbb{1}_A(X)]} \leq 2\sqrt{2\frac{\log \delta^{-1}}{n}}\sqrt{\frac{[2\log(\delta^{-1})/n] \vee (1 - \mathbb{P}[X \in A])}{\mathbb{P}[X \in A]}}.$$

Replacing $\delta$ by $\delta/2$ gives the result for $X$. The result for $Y$ follows from a similar reasoning.   $\square$

The same proof argument gives an identical result for $Y$.

**Lemma 4.** *For any $B \in \Sigma_{\mathcal{Y}}$ and any $\delta \in (0,1)$, it holds w.p.a.l. $1 - \delta$*

$$|\widehat{\mathsf{E}}_y[\mathbb{1}_B] - \mathbb{E}[\mathbb{1}_B(Y)]| \leq 2\frac{\log 2\delta^{-1}}{n} + \sqrt{\mathbb{P}[Y \in B](1 - \mathbb{P}[Y \in B])}\sqrt{2\frac{\log 2\delta^{-1}}{n}}.$$

*Assume in addition that $\mathbb{P}(Y \in B) \geq 2\sqrt{2\frac{\log 2\delta^{-1}}{n}}$. Then it holds w.p.a.l. $1 - \delta$*

$$\frac{|\widehat{\mathsf{E}}_y[\mathbb{1}_B] - \mathbb{E}[\mathbb{1}_B(Y)]|}{\mathbb{E}[\mathbb{1}_B(Y)]} \leq \sqrt{2\frac{\log 2\delta^{-1}}{n}}\sqrt{1 \vee \frac{1 - \mathbb{P}[Y \in B]}{\mathbb{P}[Y \in B]}}.$$

## B.5   Sub-Gaussian case

**Sub-Gaussian setting.**    We derive another concentration result under a less restricted sub-Gaussian condition on functions $u^\theta$ and $v^\theta$. This result relies on Pinelis and Sakhanenko's inequality for random variables in a separable Hilbert space, see (Caponnetto and De Vito, 2007, Proposition 2).

Let $\psi_2(x) = e^{x^2} - 1$, $x \geq 0$. We define the $\psi_2$-Orlicz norm of a random variable $\eta$ as

$$\|\eta\|_{\psi_2} := \inf\left\{C > 0 \ : \ \mathbb{E}\left[\psi_2\left(\frac{|\eta|}{C}\right)\right] \leq 1\right\}.$$

We recall the definition of a sub-Gaussian random vector.

**Definition 4** (Sub-Gaussian random vector). *A centered random vector $X \in \mathbb{R}^{\underline{d}}$ will be called sub-Gaussian iff, for all $u \in \mathbb{R}^{\underline{d}}$,*

$$\|\langle X, u \rangle\|_{\psi_2} \lesssim \|\langle X, u \rangle\|_{L_2(\mathbb{P})}.$$

**Proposition 1.** *Caponnetto and De Vito (2007, Proposition 2) Let $A_i$, $i \in [n]$ be i.i.d copies of a random variable $A$ in a separable Hilbert space with norm $\|\cdot\|$. If there exist constants $L > 0$ and $\sigma > 0$ such that for every $m \geq 2$, $\mathbb{E}\|A\|^m \leq \frac{1}{2}m!L^{m-2}\sigma^2$, then with probability at least $1 - \delta$*

$$\left\| \frac{1}{n} \sum_{i \in [n]} A_i - \mathbb{E}A \right\| \leq \frac{4\sqrt{2}}{\sqrt{n}} \sqrt{\sigma^2 + \frac{L^2}{n}} \log \frac{2}{\delta}. \tag{31}$$

**Lemma 5** ((Sub-Gaussian random variable) Lemma 5.5. in Vershynin (2011)). *Let $Z$ be a random variable. Then, the following assertions are equivalent with parameters $K_i > 0$ differing from each other by at most an absolute constant factor.*

1. *Tails: $\mathbb{P}\{|Z| > t\} \leq \exp(1 - t^2/K_1^2)$ for all $t \geq 0$;*

2. *Moments: $(\mathbb{E}|Z|^p)^{1/p} \leq K_2\sqrt{p}$ for all $p \geq 1$;*

3. *Super-exponential moment: $\mathbb{E}\exp(Z^2/K_3^2) \leq 2$.*

*A random variable $Z$ satisfying any of the above assertions is called a sub-Gaussian random variable. We will denote by $K_3$ the sub-Gaussian norm.*

Consequently, a sub-Gaussian random variable satisfies the following equivalence of moments property. There exists an absolute constant $c > 0$ such that for any $m \geq 2$,

$$\left(\mathbb{E}|Z|^m\right)^{1/m} \leq cK_3\sqrt{m}\left(\mathbb{E}|Z|^2\right)^{1/2}.$$

**Lemma 6.** *Assume that $\|u^\theta(X) - \mathbb{E}[u^\theta(X)]\|$ and $\|v^\theta(Y) - \mathbb{E}[v^\theta(Y)]\|$ are sub-Gaussian with sub-Gaussian norm $K$. We set $\sigma_\theta^2(X) := \mathrm{Var}(\|u^\theta(X) - \mathbb{E}[u^\theta(X)]\|)$, $\sigma_\theta^2(Y) := \mathrm{Var}(\|v^\theta(Y) - \mathbb{E}[v^\theta(Y)]\|)$. Then there exists an absolute constant $C > 0$ such that for any $\delta \in (0, 1)$, it holds w.p.a.l. $1 - \delta$*

$$\|\widehat{\mathsf{E}}_x[u^\theta] - \mathbb{E}[u^\theta(X)]\| \leq \frac{C}{\sqrt{n}} \sqrt{\sigma_\theta^2(X) + \frac{K^2}{n}} \log(2\delta^{-1}).$$

*Similarly, w.p.a.l. $1 - \delta$*

$$\|\widehat{\mathsf{E}}_y[v^\theta] - \mathbb{E}[v^\theta(Y)]\| \leq \frac{C}{\sqrt{n}} \sqrt{\sigma_\theta^2(Y) + \frac{K^2}{n}} \log(2\delta^{-1})$$

*Proof.* Set $Z := \|u^\theta(X) - \mathbb{E}u^\theta(X)\|$ and we recall that $\sigma_\theta^2(X) := \mathrm{Var}(\|u^\theta(X) - \mathbb{E}[u^\theta(X)]\|)$. We check that the moment condition,

$$\mathbb{E}Z^m \leq \frac{1}{2}m!L^{m-2}\sigma_\theta^2(X)^2, \quad \forall m \geq 2,$$

for some constant $L > 0$ to be specified.

The condition is obviously satisfied for $m = 2$. Next for any $m \geq 3$, the Cauchy-Schwarz inequality and the equivalence of moment property give

$$\mathbb{E}Z^m \leq \left(\mathbb{E}Z^{2(m-2)}\right)^{1/2} \left(\mathbb{E}Z^4\right)^{1/2} \leq 4K_3^2\sigma_\theta^2(X)^2 \left(\mathbb{E}Z^{2(m-2)}\right)^{1/2}.$$

Next, by homogeneity, rescaling $Z$ to $Z/K_1$ we can assume that $K_1 = 1$ in Lemma 5. We recall that if $Z$ is in addition non-negative random variable, then for every integer $p \geq 1$, we have

$$\mathbb{E}Z^p = \int_0^\infty \mathbb{P}\{Z \geq t\} pt^{p-1} \, dt \leq \int_0^\infty e^{1-t^2} pt^{p-1} \, dt = \left(\frac{ep}{2}\right)\Gamma\left(\frac{p}{2}\right).$$

With $p = 2(m - 2)$, we get that $\mathbb{E}Z^p \leq e(m - 2)\Gamma(m - 2) = e(m - 2)! = em!/2$. Using again Lemma 5, we can take $L = cK$ for some large enough absolute constant $c > 0$. Then Proposition 1 gives the result.

$\square$

## B.6 Estimation of conditional expectation and Conditional covariance

We now derive guarantees for the estimation of the conditional expectation and the conditional covariance for vector-valued output $Y \in \mathbb{R}^{d_y}$.

We start with a general result for arbitrary vector-valued functions of $Y$. We consider a vector-valued function $\underline{h} = (h_1, \ldots, h_{\underline{d}})$ where $h_j \in L^2_\nu(\mathcal{Y})$ for any $j \in [\underline{d}]$. We introduce the space of square integrable vector-valued functions $[L^2_\nu(\mathcal{Y}, \mathbb{R}^{\underline{d}})]$ equipped with the norm

$$\|\underline{h}\| = \sqrt{\sum_{j \in [\underline{d}]} \|h_j\|^2_{L^2_\nu(\mathcal{Y})}}.$$

Next we can define the conditional expectation of $\underline{h}(Y) = (h_1(Y^{(1)}), \ldots h_{\underline{d}}(Y^{(d_y)}))^\top$ conditionally on $X \in A$ as follows

$$\mathbb{E}[\underline{h}(Y) \mid X \in A] = \left( \mathbb{E}[h_1(Y)] + \frac{\langle \mathbb{1}_A, \mathsf{D}_{Y|X} h_1 \rangle}{\mathbb{P}(X \in A)}, \ldots, \mathbb{E}[h_{\underline{d}}(Y)] + \frac{\langle \mathbb{1}_A, \mathsf{D}_{Y|X} h_{\underline{d}} \rangle}{\mathbb{P}(X \in A)} \right)^\top$$

$$= \mathbb{E}[\underline{h}(Y)] + \frac{\langle \mathbb{1}_A, [\mathbb{1}_{\underline{d}} \otimes \mathsf{D}_{Y|X}]\underline{h} \rangle}{\mathbb{P}(X \in A)}.$$

We define similarly its empirical version as

$$\widehat{\mathsf{E}}^\theta[\underline{h}(Y) \mid X \in A] = \left( \widehat{\mathsf{E}}_y[h_1] + \frac{\langle \mathbb{1}_A, \widehat{\mathsf{D}}^\theta_{Y|X} h_1 \rangle}{\widehat{\mathsf{E}}_x[\mathbb{1}_A]}, \ldots, \widehat{\mathsf{E}}_y[h_{\underline{d}}] + \frac{\langle \mathbb{1}_A, \widehat{\mathsf{D}}^\theta_{Y|X} h_{\underline{d}} \rangle}{\widehat{\mathsf{E}}_x[\mathbb{1}_A]} \right)^\top$$

$$= \widehat{\mathsf{E}}_y[\underline{h}] + \frac{\langle \mathbb{1}_A, [\mathbb{1}_{\underline{d}} \otimes \widehat{\mathsf{D}}^\theta_{Y|X}]\underline{h} \rangle}{\widehat{\mathsf{E}}_x[\mathbb{1}_A]}.$$

Assuming that $\underline{h}(Y)$ is sub-Gaussian, we set

$$K := \|\|\underline{h}(Y) - \mathbb{E}[\underline{h}(Y)]\|\|_{\psi_2}, \quad \sigma^2(\underline{h}(Y)) := \mathrm{Var}(\|\underline{h}(Y) - \mathbb{E}[\underline{h}(Y)]\|).$$

Define

$$\underline{\psi}_n(\delta) := \frac{1}{\sqrt{n}} \sqrt{\sigma^2(\underline{h}(Y)) + \frac{K^2}{n} \log(3\delta^{-1})}$$

$$+ \frac{\|\underline{h}\|}{\sqrt{\mathbb{P}(X \in A)}} \left( \psi_n(\delta/3) + 2(1 + \psi_n(\delta/3))\varphi_X(A)\bar{\epsilon}_n(\delta/3) \right).$$

**Theorem 4.** *Let the assumptions of Theorem 2 be satisfied. Assume in addition that $\underline{h}(Y)$ is sub-Gaussian. Then we have w.p.a.l. $1 - \delta$ that*

$$\|\widehat{\mathsf{E}}^\theta[\underline{h}(Y) \mid X \in A] - \mathbb{E}[\underline{h}(Y) \mid X \in A]\| \lesssim \underline{\psi}_n(\delta). \tag{32}$$

*Proof.* We have

$$\|\mathbb{E}[\underline{h}(Y) \mid X \in A] - \widehat{\mathsf{E}}^\theta[\underline{h}(Y) | X \in A]\|$$

$$\leq \|\mathbb{E}[\underline{h}(Y)] - \widehat{\mathsf{E}}_y[\underline{h}]\| + \frac{\|\mathsf{D}_{Y|X} - \widehat{\mathsf{D}}^\theta_{Y|X}\|}{\sqrt{\mathbb{P}(X \in A)}} \|\underline{h}\| + |\langle \mathbb{1}_A, [\mathbb{1}_{\underline{d}} \otimes \widehat{\mathsf{D}}^\theta_{Y|X}]\underline{h} \rangle| \left| \frac{1}{\widehat{\mathsf{E}}_x[\mathbb{1}_A]} - \frac{1}{\mathbb{P}(X \in A)} \right|$$

$$\leq \|\mathbb{E}[\underline{h}(Y)] - \widehat{\mathsf{E}}_y[\underline{h}]\| + \frac{\|\mathsf{D}_{Y|X} - \widehat{\mathsf{D}}^\theta_{Y|X}\|}{\sqrt{\mathbb{P}(X \in A)}} \|\underline{h}\|$$

$$+ \sqrt{\mathbb{P}(X \in A)}(\|\mathsf{D}_{Y|X}\| + \|\mathsf{D}_{Y|X} - \widehat{\mathsf{D}}^\theta_{Y|X}\|)\|\underline{h}\| \left| \frac{\mathbb{P}(X \in A) - \widehat{\mathsf{E}}_x[\mathbb{1}_A]}{\widehat{\mathsf{E}}_x[\mathbb{1}_A]\mathbb{P}(X \in A)} \right|. \tag{33}$$

Recall that $\|\mathsf{D}_{Y|X}\| \leq 1$, (24) and Lemma 3. Hence, a union bound we get with w.p.a.l. $1 - 2\delta$ that

$$\|\mathbb{E}[\underline{h}(Y) \mid X \in A] - \widehat{\mathsf{E}}^\theta[\underline{h}(Y) | X \in A]\|$$

$$\leq \|\mathbb{E}[\underline{h}(Y)] - \widehat{\mathsf{E}}_y[\underline{h}]\| + \frac{\|\underline{h}\|}{\sqrt{\mathbb{P}(X \in A)}} \left( \psi_n(\delta) + 2(1 + \psi_n(\delta))\varphi_X(A)\bar{\epsilon}_n(\delta) \right). \tag{34}$$

We now handle the first term $\|\mathbb{E}[\underline{h}(Y)] - \widehat{\mathsf{E}}_y[\underline{h}]\|$. We recall that a similar quantity was already studied in Lemma 6. We can just replace $u^\theta(X)$ by $\underline{h}(Y) \in \mathbb{R}^{\underline{d}}$ to get the result since we assumed that $\underline{h}(Y)$ is sub-Gaussian. Hence there exists an absolute constant $C > 0$ such that w.p.a.l. $1 - \delta$

$$\|\mathbb{E}[\underline{h}(Y)] - \widehat{\mathsf{E}}_y[\underline{h}]\| \leq \frac{C}{\sqrt{n}} \sqrt{\sigma^2(\underline{h}(Y)) + \frac{K^2}{n} \log(2\delta^{-1})}.$$

$\square$

Actually, we can handle the conditional expectation $\mathbb{E}[Y \mid X \in A]$ in a more direct way. Set

$$\underline{\epsilon}_n(\delta) := \sqrt{\frac{\log(\delta^{-1}d_y)}{n}} \bigvee \frac{\log(\delta^{-1}d_y)}{n}.$$

**Corollary 2.** *Let the Assumptions of Theorem 2 be satisfied. Assume in addition that $Y$ is a sub-Gaussian vector. Then for any $\delta \in (0,1)$, it holds with probability at least $1 - \delta$ that*

$$\|\mathbb{E}[Y \mid X \in A] - \widehat{\mathsf{E}}^\theta[Y | X \in A]\| \lesssim \sqrt{\mathrm{tr}(\mathrm{Cov}(Y))}\underline{\epsilon}_n(\delta/3)$$
$$+ \frac{\|\underline{h}\|}{\sqrt{\mathbb{P}(X \in A)}} \left( \psi_n(\delta/3) + 2(1 + \psi_n(\delta/3))\varphi_X(A)\bar{\epsilon}_n(\delta/3) \right) =: \psi_n^{(1)}(\delta). \quad (35)$$

*Proof.* The proof of this result is identical to that of Theorem 4 up to (34). Now if we specify $\underline{h}(Y) = Y \in \mathbb{R}^{d_y}$. Then, applying Bernstein's inequality on each of the $d_y$ components of $\mathbb{E}[Y] - \overline{Y}_n$ and a union bound, we get w.p.a.l. $1 - \delta$

$$\|\mathbb{E}[Y] - \overline{Y}_n\| \lesssim \sqrt{\mathrm{tr}(\mathrm{Cov}(Y))}\sqrt{\frac{\log(\delta^{-1}d_y)}{n}} + \max_{j \in [d_y]} \|Y^{(j)}\|_{\psi_2} \frac{\log(\delta^{-1}d_y)}{n}.$$

Using again Definition 4, we obtain $\max_{j \in [d_y]} \|Y^{(j)}\|_{\psi_2} \lesssim \sqrt{\|\mathrm{Cov}(Y)\|} \leq \sqrt{\mathrm{tr}(\mathrm{Cov}(Y))}$.

It follows from the last two displays, w.p.a.l. $1 - \delta$

$$\|\mathbb{E}[Y] - \overline{Y}_n\| \lesssim \sqrt{\mathrm{tr}(\mathrm{Cov}(Y))}\underline{\epsilon}_n(\delta). \quad (36)$$

A union bound combining the previous display with (34) gives the first result. $\square$

We focus now on the conditional covariance estimation problem. We first define the conditional covariance as follows:

$$\mathrm{Cov}(Y|X \in A) = \mathrm{Cov}(Y) + \langle \mathbb{1}_A, [(\mathbb{1}_{d_y} \otimes \mathbb{1}_{d_y}) \otimes \mathsf{D}_{Y|X}]\underline{h} \otimes \underline{h}\rangle/\mathbb{P}[X \in A]$$
$$- \langle \mathbb{1}_A, [\mathbb{1}_{d_y} \otimes \mathsf{D}_{Y|X}]\underline{h}\rangle \otimes \langle \mathbb{1}_A, [\mathbb{1}_{d_y} \otimes \mathsf{D}_{Y|X}]\underline{h}\rangle/(\mathbb{P}[X \in A])^2. \quad (37)$$

Note that $\langle \mathbb{1}_A, [(\mathbb{1}_{d_y} \otimes \mathbb{1}_{d_y}) \otimes \mathsf{D}_{Y|X}]\underline{h} \otimes \underline{h}\rangle = (\langle \mathbb{1}_A, \mathsf{D}_{Y|X}h_j h_k\rangle)_{j,k \in [d_y]}$ is a $d_y \times d_y$ matrix. We obtain a similar decomposition for the estimator $\widehat{\mathrm{Cov}}^\theta(Y|X \in A)$ of the conditional covariance $\mathrm{Cov}(Y|X \in A)$ by replacing $\mathsf{D}_{Y|X}$ by $\widehat{\mathsf{D}}_{Y|X}^\theta$:

$$\widehat{\mathrm{Cov}}^\theta(Y|X \in A) := \widehat{\mathrm{Cov}}(Y) + \langle \mathbb{1}_A, [(\mathbb{1}_{d_y} \otimes \mathbb{1}_{d_y}) \otimes \widehat{\mathsf{D}}_{Y|X}^\theta]\underline{h} \otimes \underline{h}\rangle/\widehat{\mathsf{E}}_x[\mathbb{1}_A]$$
$$- \langle \mathbb{1}_A, [\mathbb{1}_{d_y} \otimes \widehat{\mathsf{D}}_{Y|X}^\theta]\underline{h}\rangle \otimes \langle \mathbb{1}_A, [\mathbb{1}_{d_y} \otimes \widehat{\mathsf{D}}_{Y|X}^\theta]\underline{h}\rangle/(\widehat{\mathsf{E}}_x[\mathbb{1}_A])^2. \quad (38)$$

We define the effective of covariance matrix $\mathrm{Cov}(Y)$ as follows:

$$\mathbf{r}(\mathrm{Cov}(Y)) := \frac{\mathrm{tr}(\mathrm{Cov}(Y))}{\|\mathrm{Cov}(Y)\|}.$$

We set for any $\delta \in (0,1)$

$$\epsilon_n^{(2)}(\delta) := \|\mathrm{Cov}(Y)\| \left( \sqrt{\frac{\mathbf{r}(\mathrm{Cov}(Y))}{n}} + \frac{\mathbf{r}(\mathrm{Cov}(Y))}{n} + \sqrt{\frac{\log(\delta^{-1})}{n}} + \frac{\log(\delta^{-1})}{n} \right), \quad (39)$$

and

$$\psi_n^{(2)}(\delta) = \epsilon_n^{(2)}(\delta) + [\psi_n(\delta/4) + 2(1 + \psi_n(\delta/4))\varphi_X(A)\bar{\epsilon}_n(\delta/4)] \frac{(\mathbb{E}[\|Y\|^2])^2}{\sqrt{\mathbb{P}[X \in A]}}$$
$$+ \psi_n^{(1)}(\delta/4) [2\|\mathbb{E}[Y \mid X \in A]\| + \psi_n^{(1)}(\delta/4)].$$

**Corollary 3.** *Let the assumptions of Corollary 2 be satisfied. Then for any $\delta \in (0,1)$, it holds with probability at least $1 - \delta$ that*

$$\|\widehat{\text{Cov}}^\theta(Y|X \in A) - \text{Cov}(Y|X \in A)\| \lesssim \psi_n^{(2)}(\delta). \tag{40}$$

*Proof.* We use again the function $\underline{h}(Y) = Y$. We note in view of (37)-(38) that

$$\|\widehat{\text{Cov}}^\theta(Y|X \in A) - \text{Cov}(Y|X \in A)\| \leq \|\widehat{\text{Cov}}(Y) - \text{Cov}(Y)\|$$

$$+ \left\|\left\langle \mathbb{1}_A, \left[(\mathbb{1}_{d_y} \otimes \mathbb{1}_{d_y}) \otimes \left(\frac{\mathsf{D}_{Y|X}}{\mathbb{P}[X \in A]} - \frac{\widehat{\mathsf{D}}_{Y|X}^\theta}{\widehat{\mathsf{E}}_x[\mathbb{1}_A]}\right)\right] \underline{h} \otimes \underline{h} \right\rangle\right\|$$

$$+ \|\mathbb{E}[\underline{h}(Y) \mid X \in A] \otimes \mathbb{E}[\underline{h}(Y) \mid X \in A] - \widehat{\mathsf{E}}^\theta[\underline{h}(Y) \mid X \in A] \otimes \widehat{\mathsf{E}}^\theta[\underline{h}(Y) \mid X \in A]\|, \tag{41}$$

Next, we note that

$$\left\|\left\langle \mathbb{1}_A, \left[(\mathbb{1}_{d_y} \otimes \mathbb{1}_{d_y}) \otimes \left(\frac{\mathsf{D}_{Y|X}}{\mathbb{P}[X \in A]} - \frac{\widehat{\mathsf{D}}_{Y|X}^\theta}{\widehat{\mathsf{E}}_x[\mathbb{1}_A]}\right)\right] \underline{h} \otimes \underline{h} \right\rangle\right\|$$

$$\leq \left\|\left\langle \mathbb{1}_A, \left[(\mathbb{1}_{d_y} \otimes \mathbb{1}_{d_y}) \otimes \left(\frac{\mathsf{D}_{Y|X}}{\mathbb{P}[X \in A]} - \frac{\widehat{\mathsf{D}}_{Y|X}^\theta}{\widehat{\mathsf{E}}_x[\mathbb{1}_A]}\right)\right] \underline{h} \otimes \underline{h} \right\rangle\right\|_{HS}$$

$$\leq \sqrt{\mathbb{P}[X \in A]} \left\|\frac{\mathsf{D}_{Y|X}}{\mathbb{P}[X \in A]} - \frac{\widehat{\mathsf{D}}_{Y|X}^\theta}{\widehat{\mathsf{E}}_x[\mathbb{1}_A]}\right\| \sum_{j,k \in [d_y]} \|Y_j Y_k\|_{L_\nu^2(\mathcal{Y})}$$

$$\lesssim \sqrt{\mathbb{P}[X \in A]} \left(\left\|\frac{\mathsf{D}_{Y|X} - \widehat{\mathsf{D}}_{Y|X}^\theta}{\mathbb{P}[X \in A]}\right\| + \|\widehat{\mathsf{D}}_{Y|X}^\theta\| \left(\frac{1}{\mathbb{P}[X \in A]} - \frac{1}{\widehat{\mathsf{E}}_x[\mathbb{1}_A]}\right)\right) \sum_{j,k \in [d_y]} \|Y_j Y_k\|_{L_\nu^2(\mathcal{Y})}$$

Remind that $Y$ is a sub-Gaussian vector. Using the equivalence of moments property of sub-Gaussian vector, we get that

$$\|Y_j Y_k\|_{L_\nu^2(\mathcal{Y})} \leq \sqrt{\mathbb{E}[Y_j^4]\mathbb{E}[Y_k^4]} \lesssim \mathbb{E}[Y_j^2]\mathbb{E}[Y_k^2], \quad \forall j, k \in [d_y].$$

By a union bound combining the last two displays with (24) and Lemma 3, we get w.p.a.l. $1 - 2\delta$

$$\left\|\left\langle \mathbb{1}_A, \left[(\mathbb{1}_{d_y} \otimes \mathbb{1}_{d_y}) \otimes \left(\frac{\mathsf{D}_{Y|X}}{\mathbb{P}[X \in A]} - \frac{\widehat{\mathsf{D}}_{Y|X}^\theta}{\widehat{\mathsf{E}}_x[\mathbb{1}_A]}\right)\right] \underline{h} \otimes \underline{h} \right\rangle\right\|$$

$$\leq [\psi_n(\delta) + 2(1 + \psi_n(\delta))\varphi_X(A)\bar{\epsilon}_n(\delta)] \frac{(\mathbb{E}[\|Y\|^2])^2}{\sqrt{\mathbb{P}[X \in A]}}. \tag{42}$$

Next, we set $u = \mathbb{E}[\underline{h}(Y) \mid X \in A]$ and $\hat{u} = \widehat{\mathsf{E}}^\theta[\underline{h}(Y) \mid X \in A]$. Then we have

$$\|u \otimes u - \hat{u} \otimes \hat{u}\| \leq \|u - \hat{u}\|(\|u\| + \|\hat{u}\|) \leq \|u - \hat{u}\|(2\|u\| + \|\hat{u} - u\|).$$

We apply next Corollary 2 to get w.p.a.l. $1 - \delta$

$$\|u \otimes u - \hat{u} \otimes \hat{u}\| \leq \psi_n^{(1)}(\delta)[2\|\mathbb{E}[Y \mid X \in A]\| + \psi_n^{(1)}(\delta)]. \tag{43}$$

Next Koltchinskii and Lounici (2017, Theorem 4) guarantees that w.p.a.l $1 - \delta$

$$\|\widehat{\text{Cov}}(Y) - \text{Cov}(Y)\| \lesssim \epsilon_n^{(2)}(\delta), \tag{44}$$

where $\epsilon_n^{(2)}(\delta)$ is defined in (39).

A union bound combining (41), (42), (43) and (44) gives the result.

$\square$

## C   Numerical Experiments

Experiments were conducted on a high-performance computing cluster equipped with an Intel(R) Xeon(R) Silver 4210 CPU @ 2.20GHz Sky Lake CPU, 377GB RAM, and an NVIDIA Tesla V100 16Gb GPU. Code is available at `https://github.com/CSML-IIT-UCL/NCP`.

### C.1   Conditional Density Estimation

To evaluate our method's ability to estimate conditional densities, we tested NCP on six different data models (described in the following paragraph) and compared its performance with ten other methods (detailed in Tab. 2). We assessed the methods' performance using the KS distance between the estimated conditional CDF and the true CDF. Additionally, we explored how the performance of each method scales with the number of training samples, ranging from $10^2$ to $10^5$, with a validation set of $10^3$ samples. We tested each method on nineteen different conditional values uniformly sampled between the 5%- and 95%-percentile of $p(x)$. Conditional CDFs were estimated on a grid of 1000 points uniformly distributed over the support of $Y$. The KS distance between each pair of CDFs was averaged over all the conditioning values. In Tab. 5, we present the mean performance (KS distance $\pm$ standard deviation), computed over 10 repetitions, each with a different random seed.

**Synthetic data models.**   We included the following synthetic datasets from Rothfuss et al. (2019) and Gao and Hastie (2022) into our benchmark:

- `LinearGaussian`, a simple univariate linear density model defined as $Y = X + \mathcal{N}(0, 0.1)$ where $X \sim \text{Unif}(-1, 1)$.

- `EconDensity`, an economically inspired heteroscedastic density model with a quadratic dependence on the conditional variable defined as $Y = X^2 + \epsilon_Y, \epsilon_Y \sim \mathcal{N}(0, 1 + X)$ where $X \sim |\mathcal{N}(0, 1)|$.

- `ArmaJump`, a first-order autoregressive model with a jump component exhibiting negative skewness and excess kurtosis, defined as

$$x_t = [c(1 - \alpha) + \alpha x_{t-1}] + (1 - z_t)\epsilon_t + z_t [-3c + 2\epsilon_t],$$

  where $\epsilon_t \sim \mathcal{N}(0, 0.05)$ and $z_t \sim B(1, p)$ denote a Gaussian shock and a Bernoulli distributed jump indicator with probability $p$, respectively. The parameters were left at their default value.

- `GaussianMixture`, a bivariate Gaussian mixture model with 5 kernels where the goal is to estimate the conditional density of one variable given the other. The mixture model is defined as $p(X, Y) = \sum_{k=1}^{5} \pi_k \mathcal{N}(\boldsymbol{\mu_k}, \boldsymbol{\Sigma_k})$ where $\pi_k$, $\boldsymbol{\mu_k}$, and $\boldsymbol{\Sigma_k}$ are the mixing coefficient, mean vector, and covariance matrix of the $k$-th distribution. All the parameters were randomly initialized.

- `SkewNormal`, a univariate skew normal distribution defined as $Y = 2\phi(X)\psi(\alpha X)$ where $\phi(\cdot)$ and $\psi(\cdot)$ are the standard normal probability and cumulative density functions, and $\alpha$ is a parameter regulating the skewness. The parameters were left at their default value.

- Locally Gaussian or Gaussian mixture distribution (LGGMD) (Gao and Hastie, 2022), a regression dataset where the target $y$ depends on the three first dimensions of $x$, with seventeen irrelevant features added to $x$. The features of $x$ are all uniformly distributed between $-1$ and $1$. The first dimension of $x$ gives the mean of $Y|X$, the second is whether the data is Gaussian or a mixture of two Gaussians, and the third gives its asymmetry. More specifically:

$$Y|X \sim \begin{cases} 0.5\mathcal{N}(0.25X^{(1)} - 0.5, 0.25(0.25X^{(3)} + 0.5)^2) \\ \quad +0.5\mathcal{N}(0.25X^{(1)} + 0.5, 0.25(0.25X^{(3)} - 0.5)^2) \text{ if } X^{(2)} \leq 0.2 \\ 0\mathcal{N}(0.25X^{(1)} - 0.5, 0.3) \text{ if } X^{(2)} > 0.2 \end{cases} \quad (45)$$

To sample data from `EconDensity`, `ArmaJump`, `GaussianMixture`, and `SkewNormal`, we used the library `Conditional_Density_Estimation` (Rothfuss et al., 2019) available at `https://github.com/freelunchtheorem/Conditional_Density_Estimation`.

**Training NCP.** We trained an NCP model with $u^\theta$ and $v^\theta$ as multi-layer perceptrons (MLPs), each having two hidden layers of 64 units using GELU activation function in between. The vector $\sigma^\theta$ has a size of $d = 100$, and $\gamma$ is set to $10^{-3}$. Optimization was performed over $10^4$ epochs using the Adam optimizer with a learning rate of $10^{-3}$. Early stopping was applied based on the validation set with patience of 1000 epochs. To ensure the positiveness of the singular values, we transform the vector $\sigma^\theta$ with the Gaussian function $x \mapsto \exp(-x^2)$ during any call of the forward method. Whitening was applied at the end of training.

**Compared methods.** We compared our NCP network with ten different CDE methods. See Tab. 2 for the exhaustive list of models including a brief summary and key hyperparameters.

In particular, the methods were set up as follows:

- NF was characterized by a $1D$ Gaussian base distribution and two Masked Affine Autoregressive flows (Papamakarios et al., 2017) followed by a LU Linear permutation flow. To match the NCP architecture, each flow was defined by two hidden layers with 64 units each. The training procedure was the same as for the NCP model. The model was implemented using the library `normflows` (Stimper et al., 2023).

- DDPM was characterized by a U-Net (Ronneberger et al., 2015), a noise schedule starting from $10^-4$ to $0.02$ and 400 steps of diffusion as implemented in `https://github.com/TeaPearce/Conditional_Diffusion_MNIST`.

- CKDE's kernels bandwidth was estimated according to Silverman's rule (Silverman, 1986).

- MDN's architecture was defined by two hidden layers with 64 units each and 20 Gaussians kernels.

- KMN's architecture was defined by two hidden layers with 64 units each, 50 Gaussians kernels, and kernels bandwidth was estimated according to Silverman's rule (Silverman, 1986).

- LSCDE was defined by 500 components which bandwiths were set to 0.5 and kernels center found via a k-means procedure.

- NNKDE's number of neighbors was set using the heuristics $k = \sqrt{n}$ (Devroye et al., 1996). Kernels bandwidth was estimated according to Silverman's rule (Silverman, 1986). We used the implementation available at `https://github.com/lee-group-cmu/NNKCDE`.

- RFCDE was characterized by a Random Forest with 1000 trees and 31 cosine basis functions. The training was performed using the `rfcde` library available at `https://github.com/lee-group-cmu/RFCDE`.

- FC was trained using a Random Forest with 1000 trees as a regression method and had 31 cosine basis functions. The training was performed using the `flexcode` library available at `https://github.com/lee-group-cmu/FlexCode`.

- LinCDE was trained with 1000 LinCDE trees using the `LinCDE.boost` R function from `https://github.com/ZijunGao/LinCDE`.

CKDE, MDN, KMN, and LSCDE hyperparameters were set according to Rothfuss et al. (2019) and were trained using the library `Conditional_Density_Estimation` available at `https://github.com/freelunchtheorem/Conditional_Density_Estimation`. All methods involving the training of a neural network were assigned the same number of epochs given to NCP. All other method parameters were set as prescribed in their paper.

**Results.** See Tab. 4 for the comparison of performances for $n = 10^4$. See also Fig. 5. We also carried out an ablation study on centering and whitening post-treatment for NCP in Tab. 3

### C.2 Confidence Regions

The objective of this next experiment is to estimate a confidence interval at coverage level $90\%$ for two distribution models with different properties (Laplace and Cauchy) and one real dataset in order to showcase the versatility of our NCP approach.

Table 2: Compared methods for the CDE problem.

| Method | Summary | Main hyperparams |
|---|---|---|
| Normalizing Flows (NF) (Rezende and Mohamed, 2015b) | Generative models that transform a simple distribution into a complex one through a series of invertible and differentiable transformations | • Architecture 
 • Flow type |
| Denoising Diffusion Probabilistic Model (DDPM) (Ho et al., 2020) | Generative models that learn to generate data by reversing a gradual noising process, modeling distributions through iterative refinement | • Number of diffusion steps 
 • Noise schedule |
| Conditional KDE (CKDE) (Li and Racine, 2006) | Nonparametric approach modeling the joint and marginal probabilities via KDE and computes the conditional density as $p(y\|x) = p(x,y)/p(x)$. | • KDE bandwidth |
| Mixture Density Network (MDN) (Bishop, 1994) | Uses NeuralNets which takes conditional $x$ as input and governs all the weights of a GMM modeling $p(y\|x)$. | • NeuralNet architecture 
 • Number of kernels |
| Kernel Mixture Network (KMN) (Ambrogioni et al., 2017) | Similar to MDN with the difference that NN only controls the weights of the GMM. | • NeuralNet architecture 
 • Method for finding kernel centers 
 • Number of kernels |
| Least-Squares CDE (LSCDE) (Sugiyama et al., 2010) | Computes the conditional density as linear combination of Gaussian kernels | • Method for finding kernel centers 
 • Number of kernels 
 • Kernels' bandwidth |
| Nearest Neighbor Kernel CDE (NNKDE) (Izbicki et al., 2017) (Freeman et al., 2017) | Uses nearest neighbors of the evaluation point $x$ to compute a KDE estimation of $y$. | • Number of neighbors 
 • Kernel bandwidth |
| Random Forest CDE (RFCDE) (Pospisil and Lee, 2018) | Uses a random forest to partition the feature space and constructs a weighted KDE of the output space, based on the weights of the leaves in the forest. | • Random forest hyperparams 
 • Basis system 
 • Number of basis |
| Flexible CDE (FC) (Izbicki and Lee, 2017) | Nonparametric approach which uses a basis expansion of univariate $y$ to turn CDE into a series of univariate regression problems. | • Number of expansion coeffs 
 • Regression method hyperparams. |
| LinCDE (LCDE) (Gao and Hastie, 2022) | Conditional training of unconditional machine learning models to learn density | • Number of LinCDE trees |

Table 3: Ablation study on post-treatment for NCP. We report the mean and std of KS distance of estimated CDF from the truth averaged over 10 repetitions with $n = 10^5$ (best method in bold red). NCP–C and NCP–W refer to our method with centering and whitening post-treatment, respectively.

| Model | LinearGaussian | EconDensity | ArmaJump | SkewNormal | GaussianMixture | LGGMD |
|---|---|---|---|---|---|---|
| NCP | $0.040 \pm 0.007$ | $0.014 \pm 0.003$ | $0.046 \pm 0.012$ | $0.023 \pm 0.006$ | $0.027 \pm 0.008$ | $0.055 \pm 0.010$ |
| NCP–C | $0.019 \pm 0.006$ | $0.010 \pm 0.003$ | $0.037 \pm 0.011$ | $0.015 \pm 0.004$ | **0.015** $\pm$ **0.004** | $0.048 \pm 0.007$ |
| NCP–W | **0.010** $\pm$ **0.000** | **0.005** $\pm$ **0.001** | **0.010** $\pm$ **0.002** | **0.008** $\pm$ **0.001** | **0.015** $\pm$ **0.004** | **0.047** $\pm$ **0.005** |

Table 4: Mean and standard deviation of KS distance of estimated CDF from the truth averaged over 10 repetitions with sample size of $10^4$ (best method in bold red, second best in bold black). NCP–C and NCP–W refer to our method with centering and whitening post-treatment, respectively.

| Model | LinearGaussian | EconDensity | ArmaJump | SkewNormal | GaussianMixture | LGGMD |
|---|---|---|---|---|---|---|
| NCP | $0.046 \pm 0.011$ | $0.021 \pm 0.009$ | $0.048 \pm 0.009$ | $0.043 \pm 0.029$ | $0.035 \pm 0.004$ | $0.188 \pm 0.011$ |
| NCP–C | $0.031 \pm 0.008$ | $0.019 \pm 0.008$ | **0.038** $\pm$ **0.011** | **0.031** $\pm$ **0.013** | **0.031** $\pm$ **0.003** | $0.189 \pm 0.012$ |
| NCP–W | **0.026** $\pm$ **0.002** | **0.016** $\pm$ **0.003** | **0.020** $\pm$ **0.002** | **0.024** $\pm$ **0.011** | **0.030** $\pm$ **0.002** | $0.176 \pm 0.014$ |
| DDPM | $0.414 \pm 0.341$ | $0.264 \pm 0.240$ | $0.358 \pm 0.314$ | $0.284 \pm 0.251$ | $0.416 \pm 0.242$ | $0.423 \pm 0.223$ |
| NF | **0.011** $\pm$ **0.002** | **0.015** $\pm$ **0.003** | $0.141 \pm 0.005$ | $0.039 \pm 0.005$ | $0.113 \pm 0.006$ | $0.288 \pm 0.010$ |
| KMN | $0.599 \pm 0.003$ | $0.349 \pm 0.019$ | $0.490 \pm 0.007$ | $0.380 \pm 0.009$ | $0.306 \pm 0.003$ | $0.225 \pm 0.008$ |
| MDN | $0.245 \pm 0.011$ | $0.051 \pm 0.002$ | $0.164 \pm 0.005$ | $0.089 \pm 0.002$ | $0.144 \pm 0.009$ | $0.232 \pm 0.008$ |
| LSCDE | $0.418 \pm 0.003$ | $0.119 \pm 0.004$ | $0.250 \pm 0.007$ | $0.109 \pm 0.002$ | $0.201 \pm 0.005$ | $0.295 \pm 0.034$ |
| CKDE | $0.187 \pm 0.001$ | $0.023 \pm 0.003$ | $0.125 \pm 0.002$ | $0.046 \pm 0.001$ | $0.085 \pm 0.003$ | $0.241 \pm 0.021$ |
| NNKCDE | $0.090 \pm 0.002$ | $0.060 \pm 0.006$ | $0.063 \pm 0.006$ | $0.052 \pm 0.005$ | $0.059 \pm 0.004$ | $0.207 \pm 0.013$ |
| RFCDE | $0.132 \pm 0.009$ | $0.136 \pm 0.010$ | $0.130 \pm 0.009$ | $0.139 \pm 0.009$ | $0.134 \pm 0.012$ | $0.162 \pm 0.006$ |
| FC | $0.090 \pm 0.004$ | $0.030 \pm 0.006$ | $0.042 \pm 0.003$ | $0.033 \pm 0.002$ | $0.033 \pm 0.003$ | **0.065** $\pm$ **0.008** |
| LCDE | $0.122 \pm 0.002$ | $0.029 \pm 0.003$ | $0.118 \pm 0.003$ | $0.064 \pm 0.007$ | $0.050 \pm 0.002$ | **0.141** $\pm$ **0.004** |

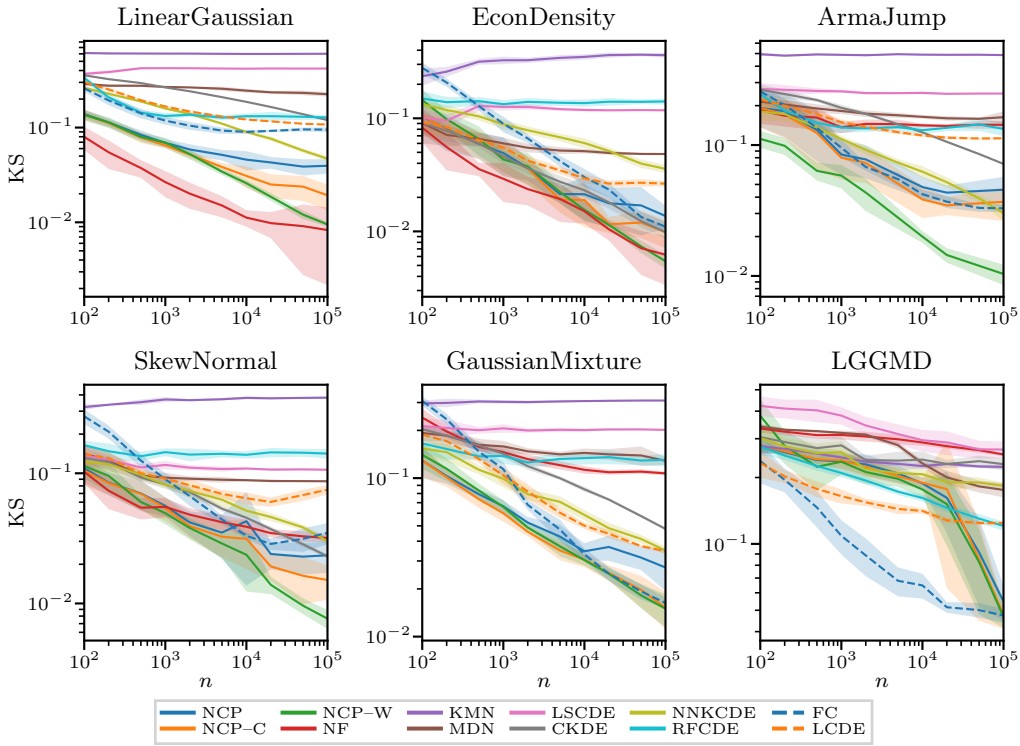

Figure 5: **Performances for CDE on synthetic datasets w.r.t sample size** $n$. Performance metric is Kolmogorov-Smirnov (KS) distance to truth.

**Compared methods.** We compared our NCP procedure for building conditional confidence intervals to the state-of-the-art conditional conformal prediction method in Gibbs et al. (2023). We also developed another method based on Normalizing Flows' estimation of the conditional CDE and we added it to the benchmark.

**Experiment for Laplace and Cauchy distributions.** We generate a dataset where the $X$ variable follows a uniform distribution on interval $[0, 5]$ and $Y|X = x$ follows either a Laplace distribution with location and scale parameters $(\mu(x), b(x)) = (x^2, x)$ or a Cauchy distribution with location and scale parameters $(\mu(x), b(x)) = (x^2, 1 + x)$. We create a train set of 50000 samples, a validation set of 1000 samples and a test set of 1000 samples.

For the Laplace distribution, we train an NCP where $u^\theta$ and $v^\theta$ are multi-layer perceptrons with two hidden layers of 128 cells, $\sigma^\theta$ is a vector of size $d = 500$ and $\gamma = 10^{-2}$. Between each layer, we use the GELU activation function. We optimize over 5000 epochs using the Adam optimizer with a learning rate of $10^{-3}$. We apply early stopping with regard to the validation set with a patience of 100 epochs. Whitening is applied at the end of training. To fit the Cauchy distribution, we increase the depth of the MLPs to 5 and the width to 258.

We compare this NCP network with two state-of-the-art methods. The first is a normalizing flow with base distribution a $1D$ Gaussian and two Autoregressive Rational Quadratic spline flows (Durkan et al., 2019) followed by a LU Linear permutation flow. All flows come from the library `normflows` (Stimper et al., 2023). The spline flows have each two blocks of 128 hidden units to match the NCP architecture. The normalizing flow is allowed the same number of epochs as ours with the same optimizer. The second model is the conditional conformal predictor from Gibbs et al. (2023). This model needs a regressor as an input. We consider a situation favorable to Gibbs et al. (2023) as we assume as prior knowledge that the true conditional expectation is a polynomial function (the truth is actually the quadratic function in this example). Therefore we chose a linear regression with polynomial features as in Gibbs et al. (2023) as this regressor should fit the data without any problem. For all other choices of parameters, we follow the prescriptions of Gibbs et al. (2023). For the sake of

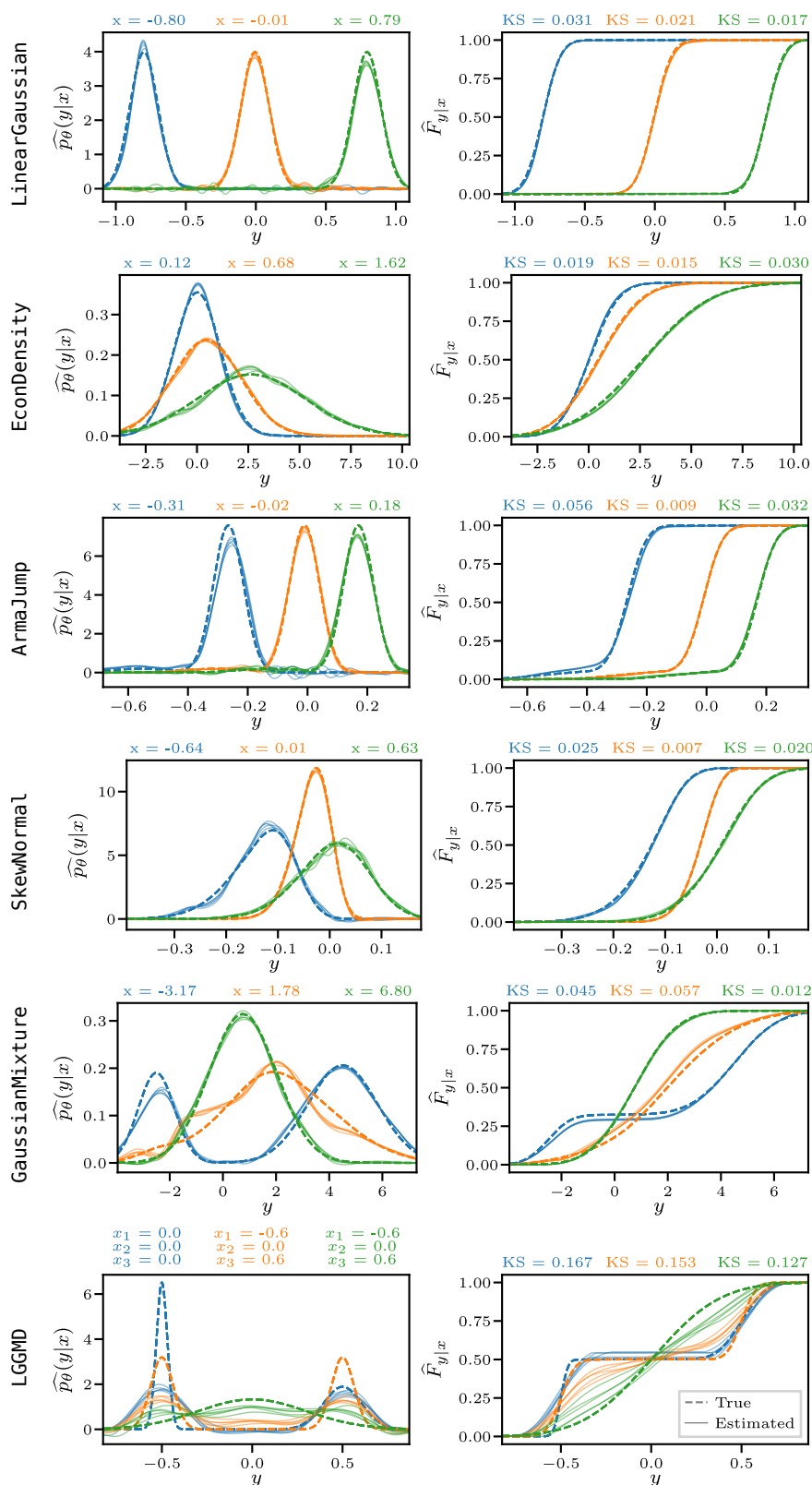

Figure 6: **Estimated conditional PDFs (left) and CDFs (right) for each synthetic dataset for 3 different conditioning points.** Dotted lines represent the true distributions, while solid lines represent the estimates from NCP. The average KS distance over 5 repetitions is also reported on the right plots.

fairness, we note that the validation set used for early stopping in NF and NCP was also used as a calibration set for the CCP method.

By design, the Conditional Conformal Predictor (CCP) gives the confidence interval directly. However NCP and NF output the conditional distribution. To find the smallest confidence interval with desired coverage, we apply the linear search algorithm described in Algorithm 3 on the discretized conditional CDFs provided by NCP and NF. The results are provided in Fig. 1. First, observe that although the linear regression achieves the best estimation of the conditional mean, as should be expected since the model is well-specified in this case, the confidence intervals, however, are unreliable for most of the considered conditioning. We also notice instability for NF and CCP for conditioning in the neighborhood of $x = 0$ with NF confidence region exploding at $x = 0$. We expect this behavior is due to the fact that the conditional distribution at $x = 0$ is degenerate. Comparatively, NCP does not exhibit such instability around $x = 0$. It only tends to overestimate the produced confidence region for conditioning close to $x = 0$.

---

**Algorithm 3** Confidence interval search given a CDF

---

**Require:** $Y$ a vector of values, $F_Y$ a vector of realisations of the CDF at points $Y$, $\alpha \in [0, 1]$ a
  confidence level
  Initialize $t_{\text{low}} = 0$ and $t_{\text{high}} = 1$
  Initialize $t_{\text{low}}^* = 0$ and $t_{\text{high}}^* = -1$
  Initialize $s* = \infty$
  **while** Center and scale $X_{\text{train}}$ and $Y_{\text{train}}$ **do**
    **if** $F_Y[t_{\text{high}}] - F_Y[t_{\text{low}}] \geq \alpha$ **then**
      size $= Y[t_{\text{high}}] - Y[t_{\text{low}}]$
      **if** size $< s*$ **then**
        $t_{\text{low}}^* = t_{\text{low}}, \ \ t_{\text{high}}^* = t_{\text{high}}, \ \ s* =$ size
      **end if**
      $t_{\text{low}} = t_{\text{low}} + 1$
    **else if** $t_{\text{high}} = \text{len}(Y) - 1$ **then**
      break
    **else**
      $t_{\text{high}} = t_{\text{high}} + 1$
    **end if**
  **end while**
  Return $Y[t_{\text{low}}], Y[t_{\text{high}}]$

---

**Experiment on real data.** We also evaluate the performance of NCP in estimating confidence intervals using the Student Performance dataset available at `https://www.kaggle.com/datasets/nikhil7280/student-performance-multiple-linear-regression/data`. This dataset comprises 10000 records, each defined by five predictors: hours studied, previous scores, extracurricular activities, sleep hours, and sample question papers practiced, with a performance index as the target variable. In this experiment, the NCP's $u^\theta$ and $v^\theta$ are defined by MLPs with two hidden layers, each containing 32 units and using GELU activation functions, $\sigma^\theta$ is a vector of size $d = 50$ and $\gamma = 10^{-2}$. Optimization was performed over 50000 epochs using the Adam optimizer with a learning rate of $10^{-3}$. We compare NCP with a normalizing flow defined as above in which spline flows have each two blocks of 32 hidden units to match NCP architecture. The normalizing flow is trained for the same number of epochs as our model, using the same optimizer. We further compare NCP with a split conformal predictor featuring a Random Forest regressor (RFSCP) with 100 estimators. We used the implementation of the library `puncc` (Mendil et al., 2023). For NCP and the normalizing flow, early stopping is based on the validation set, while for RFSCP, the validation set serves as the calibration set. We performed 10 repetitions, randomly splitting the dataset into a training set of 8000 samples and validation and test sets of 1000 samples each. We report the results of the estimated confidence interval at a coverage level of 90% in Tab. 5. The methods provide fairly good coverage. NF did not respect the 90% coverage condition. Only NCP and RFSCP both respect the coverage condition but the width of the confidence intervals for RFSCP are larger than for NCP.

**Discussion on Conformal Prediction.** Conformal prediction (CP) is a popular model-agnostic framework for uncertainty quantification approach Vovk et al. (1999). CP assigns nonconformity

Table 5: Mean and standard deviation of 90% prediction interval (PI) coverages and interval widths, averaged over 10 repetitions for the Student Performance dataset from Kaggle. NCP–C and NCP–W refer to our method with centering and whitening post-treatment, respectively.

| Model | Coverage 90% PI | Width 90% PI |
|---|---|---|
| NCP–C | $89.41\%_{\ \pm\ 2.12\%}$ | $0.39_{\ \pm\ 0.02}$ |
| NCP–W | $91.02\%_{\ \pm\ 0.72\%}$ | $0.38_{\ \pm\ 0.01}$ |
| NF | $89.10\%_{\ \pm\ 1.07\%}$ | $0.35_{\ \pm\ 0.00}$ |
| RFSCP | $90.03\%_{\ \pm\ 1.06\%}$ | $0.41_{\ \pm\ 0.01}$ |

scores to new data points. These scores reflect how well each point aligns with the model's predictions. CP then uses these scores to construct a prediction region that guarantees the true outcome will fall within it with a user-specified confidence parameter. However, CP is not without limitations. The construction of these guaranteed prediction regions can be computationally expensive especially for large datasets, and need to be recomputed from scratch for each value of the confidence level parameter. In addition, the produced CP confidence regions tend to be conservative. Another limitation of regular CP is that predictions are made based on the entire input space without considering potential dependencies between variables. Conditional conformal prediction (CCP) was later developed to handle conditional dependencies between variables, allowing in principle for more accurate and reliable predictions Gibbs et al. (2023). CCP suffers from the typical limitations of regular CP and the theoretical guarantees.

### C.3 High-dimensional Experiments

**Experiment on high-dimensional synthetic data.** In Fig. 3, we trained NCP for $d = 100$ using the same MLP architecture and the same NF with autoregressive flow as in our initial experiments based on $n = 10^5$ samples $\{(X_i, Y_i)\}_{i=1}^n$ with values in $\mathbb{R}^d \times \mathcal{Y}$. We plot the conditional CDF for several conditioning w.r.t. $\theta(x)$ on 10 repetitions. NCP paired with a small MLP architecture performs comparably to the NF model for Gaussian distributions. For discrete distributions, the NCP demonstrates superior performance compared to the NF model.

We repeated the experiment in Fig. 3 for $d \in \{100, 200, 500, 1000\}$ and recorded the average Kolmogorov-Smirnov (KS) distance of the NCP conditional distribution to the truth, computation time and their standard deviations over 10 repetitions.

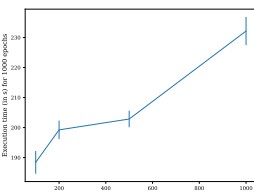

| $\theta$ | 100 | | 500 | | 1000 | |
|---|---|---|---|---|---|---|
| | mean | std | mean | std | mean | std |
| 1.0 | 0.057 | 0.019 | 0.079 | 0.050 | 0.062 | 0.016 |
| 1.57 | 0.041 | 0.009 | 0.069 | 0.042 | 0.049 | 0.017 |
| 3.14 | 0.030 | 0.016 | 0.116 | 0.173 | 0.036 | 0.010 |
| 5.0 | 0.067 | 0.052 | 0.131 | 0.175 | 0.072 | 0.036 |

Figure 7: **Left:** we observe only $\approx 20\%$ increase in compute time going from $d = 10^2$ to $d = 10^3$. **Right:** average KS distance to the truth and standard deviation over 10 repetitions.

**High-dimensional experiment in molecular dynamics: Chignolin folding.** We investigated the dynamics of Chignolin folding, using a molecular dynamics simulation lasting $106\mu s$ and sampled every $200ps$, resulting in $524, 743$ data points. Our analysis focuses on 39 heavy atoms (nodes) with a cutoff radius of 5 Angstroms. To predict the conditional transition probability between metastable states, we integrate our NCP approach with a graph neural network (GNN) model. GNNs, as demonstrated by Chanussot et al. (2021), represent the state-of-the-art in modeling atomistic systems, adeptly incorporating the roto-translational and permutational symmetries inherent in physical systems. In particular, we employed a SchNet model Schütt et al. (2019, 2023) with three interaction blocks. Each block features a 64-dimensional latent atomic environment, and the inter-atomic distances for message passing are expanded over 20 radial basis functions. After the final interaction block, each latent atomic environment is processed through a linear layer and then aggregated by averaging. The model underwent training for 100 epochs using an Adam optimizer

with a learning rate of $10^{-3}$. We employed a batch size of $256$ and set $\gamma$ to $10^{-3}$. In Fig. 2, we show how our NCP approach enables the tracking transitions between metastable states, demonstrating accurate forecasting and strong uncertainty quantification.

