# OpenReview forum: "Neural Conditional Probability for Uncertainty Quantification"
_NeurIPS.cc/2024/Conference — NeurIPS 2024 poster_

### Official Review · Reviewer_6sLX · 2024-07-09

**Soundness:** 3
**Presentation:** 3
**Contribution:** 3
**Rating:** 7
**Confidence:** 2

**Summary:**

This paper proposes Neural Conditional Probability (NCP), a novel operator-theoretic approach for learning conditional probability distributions. Extensive theoretical results are provided to support the optimization consistency and statistical accuracy of NCP. NCP can be used to extract conditional density and compute statistical measures such as conditional mean, variance, moments and CDF once it is trained. Experiments on a collection of conditional density estimation datasets are conducted to highlight the efficacy of NCP.

**Strengths:**

- This paper is mathematically solid and well-organized.
- This paper focuses on a fundamental problem of learning conditional distribution in statistical learning and introduces an effective and simplistic approach that outperforms baselines with more complex architectures.

**Weaknesses:**

- The proposed NCP method is not clearly motivated or introduced. In Line 49-50, the authors mention that NCP does not belong to any of the four aforementioned approaches. But how is NCP in contrast with them and in what aspects does NCP make improvements? I believe adding some intuitive explanations accompanying theoretical analysis would help improve the readability.
- Some key concepts or methods are not clearly explained, which makes it hard to understand the contributions of this work. For example, why is learning *conditional expectation operator* considered useful? Are there any baseline methods that also learn expectation operators?

**Questions:**

Please see Weaknesses.

**Limitations:**

The authors have discussed the limitations of their work.

---

> ### Author Rebuttal · Authors · 2024-08-07
>
> We appreciate the reviewer's insightful evaluation and valuable comments. In what follows, we aim to address the highlighted weaknesses and respond to the reviewer's questions.
>
> ## Weaknesses
>
> - __W1:__ We thank the reviewer for this comment. __We have added a detailed comparison of the NCP method to the main existing strategies in the global response__. In brief, while the operator approach is fundamentally different, it can be viewed as a significant improvement over direct learning strategies that rely on a pre-specified dictionary of functions. The NCP method, in contrast, learns the latent space representation (dictionary) that is best adapted to the data under investigation.
>
> - __W2:__ The conditional operator approach has been the core idea in the ML theory of kernel mean embeddings and conditional mean embeddings [A], and, more recently, dynamical systems [B]. There, the operator is estimated on a universal (infinite-dimensional) reproducing kernel Hilbert space, in order to transfer probability distributions of one marginal to another by encoding them into the mean of the canonical feature map. While this theory is rich, it is __essentially limited to infinite-dimensional feature spaces__, and, hence, suffers from scalability limitations and the need for experts to design specific kernels for application at hand. __How to build finite-dimensional data representations and inference models__ based on them, to the best of our knowledge, was __an open problem__. So, we hope that our  work on the operator approach to inference of conditional probabilities opens a new line of research into this promising methodology that offers several significant benefits. From a probabilistic perspective, our approach allows us to learn the true joint distribution of $(X,Y)$ without needing to specify a model, even when the relationship between $X$ and $Y$ is complex. This capability provides valuable insights into $X$ and $Y$. For instance, the independence of $X$ and $Y$ is conveniently captured by the nullity of the conditional expectation operator. Moreover, the conditional expectation operator enables us to derive essential objects such as the conditional PDF, conditional CDF, conditional quantiles, and moments, even in complex nonlinear scenarios, without prior knowledge about $(X,Y)$. From a computational standpoint, we utilize the spectral properties of compact operators along with the recent NCP self-supervised loss function to create a theoretically sound training loss. This loss function guarantees that the true target operator is the unique global minimizer (see Theorem 1), while being inexpensive to evaluate, facilitating fast and stable training.
>
> ### References:
> [A] Krikamol Muandet, Kenji Fukumizu, Bharath Sriperumbudur and Bernhard Schölkopf (2017), "Kernel Mean Embedding of Distributions: A Review and Beyond", Foundations and Trends® in Machine Learning: Vol. 10: No. 1-2, pp 1-141
>
> [B] Kostic, V., Novelli, P., Maurer, A., Ciliberto, C., Rosasco, L., and Pontil, M. (2022). Learning dynamical systems via Koopman operator regression in reproducing kernel Hilbert spaces. In Advances in Neural Information Processing Systems.

---

> ### Comment · Reviewer_6sLX · 2024-08-13
>
> Thank you for the detailed response. I am happy to keep my score.

---

> > ### Author Response · Authors · 2024-08-13
> > **Acknowledgement to the reviewer**
> >
> > We would like to thank the reviewer for their comments, which inspired us to make additional steps and improve our work. We are happy that our rebuttal was helpful, and we commit to incorporate it in the revised manuscript.

---

### Official Review · Reviewer_vE36 · 2024-07-09

**Soundness:** 3
**Presentation:** 3
**Contribution:** 3
**Rating:** 7
**Confidence:** 1

**Summary:**

I am not qualified to review this paper

**Strengths:**

I am not qualified to review this paper

**Weaknesses:**

I am not qualified to review this paper

**Questions:**

I am not qualified to review this paper

**Limitations:**

I am not qualified to review this paper

---

> ### Author Rebuttal · Authors · 2024-08-06
>
> We outline our contributions and offer further context in the global response, hoping these additions will better highlight the value of our work on the  inference of conditional probability and uncertainty quantification.

---

### Official Review · Reviewer_BaSA · 2024-07-09

**Soundness:** 4
**Presentation:** 3
**Contribution:** 4
**Rating:** 7
**Confidence:** 3

**Summary:**

The authors propose a method (Neural Conditional Probability, NCP) for learning a conditional distribution P(Y | X) from a finite sample from a distribution. The method is based on following observations: (1) it is sufficient to learn the conditional expectation operator E_{Y | X}[f](x) = E[f(Y) | X = x]; (2) the conditional expectation operator can be written as (an infinite) SVD decomposition which could be truncated at some point, so the problem reduced to learning the finite number of functions in the SVD decomposition; (3) the joint distribution density can be written using the functions from the SVD decomposition of the conditional expectation operator, which gives an optimisation objective for fitting the functions from the SVD decomposition using the sample from a joint distribution. The authors provide an extensive theoretical analysis of the proposed method as well as a simulation study on a few synthetic datasets.

**Strengths:**

+ An interesting, novel and theoretically well-motivated method addressing an important problem of conditional distribution estimation
+ The method uses a fairly simple neural network (MLP) but achieves the competitive to the methods using much more complex architectures
+ Thorough theoretical analysis on statistical properties of the proposed estimator

**Weaknesses:**

- Limited experiments restricted to synthetic data making it difficult to judge the potential applicability of this method
- It would be nice to have a short summary on the main properties of operators, their SVD decompositions, etc. I could generally follow the presentation without major problems, but having a such an operators summary would have made it easier to read the paper

**Questions:**

- I am wondering about the choice of the specific loss function in Eq. (6). Could, for example, the log-likelihood potentially be used here? If so, what are the advantages of using the Eq. (6) instead of log-likelihood?
- The loss function in Eq. (9) is roughly speaking a regularisation term enforcing that the singular functions are orthonormal, is it correct? Could it be possible to build the neural nets with such properties by construction rather than enforce it by regularisation?
- What do you think about the scalability of the model to more complex datasets than in Section 6? For example, conditional image generation. Do you expect issues applying NCP in such cases?

**Limitations:**

The limitations are sufficiently addressed (NeurIPS Paper Checklist)

---

> ### Author Rebuttal · Authors · 2024-08-07
>
> We appreciate the reviewer's insightful evaluation and valuable comments. In what follows, we aim to address the highlighted weaknesses and respond to the reviewer's questions.
>
> ## Weaknesses
>
> - __W1.__ Thank you for this remark. __We added several high-dimensional experiments focused on UQ tasks, while most are synthetic to explore, as requested, scalability of NCP, one is a very complex real-world problem of protein folding__, see also our global response. For the latter, we used a dataset that is considered as challenging for the task of inferring dynamics of the stochastic process governed by the conditional probability kernel. Applying NCP to this setting, we were able to infer this conditional density from learned graph representations of Chignolin mini-protein ensemble. This allowed us to predict the expected average pairwise atomic distances between heavy atoms and infer its 10th and 90th percentiles. As reported in Fig. 3, folded/unfolded protein corresponds to small/large coverage that originates from less/more variations of the ensemble. To the best of our knowledge, this is the first work that was able to perform UQ to infer metastable states of Chignolin, an important contribution in its own right.
>
> - __W2.__ We thank the referee for the suggestion. We will add a section in Appendix summarizing the fundamental properties of operator theory that we used to develop the NCP approach and refer to it at the beginning of Section 3. This reminder will include in particular the definition of operators on infinite-dimensional Hilbert spaces and the definition of compact operators. Finally we will state the Erchart-Young-Mirsky theorem which guarantees the existence of the SVD for compact operators.
>
> ## Questions
>
> - __Q1:__ We thank the referee for the interesting question. Maximizing the log-likelihood function to find model parameters is a central method in statistics, offering strong theoretical guarantees for models satisfying proper regularity conditions. However, this method becomes computationally complex for deep learning models, as the log-likelihood function can be non-concave with multiple local maxima, making it challenging to find the global maximum. In contrast, our approach represents the joint distribution through an operator and uses the regularized NCP loss designed according to the best low-rank rank approximation for operators. This method has a strong theoretical guarantee, as proven in our Theorem 1, that the unique global minimizer of this loss is the true target operator. Additionally, the empirical NCP loss is smooth and can be evaluated with linear complexity in batch size and latent space dimension, making it compatible with standard deep learning training methods.
>
> -  __Q2:__ Another great question. We've been investigating an alternative to regularization that leverages the Singular Value Decomposition (SVD) properties of compact operators to encode orthogonality directly into the model architecture, rather than enforcing it through regularization. This approach is still under investigation, and it is not yet clear if it will surpass our current regularization method. Indeed, our current regularization approach offers two main advantages:
>     -  __Ease of Computation.__ The regularization term is straightforward to compute and integrates seamlessly into the loss function, ensuring scalability and full compatibility with mini-batching and existing optimization methods.
>     - __Theoretical Guarantees.__ As established in Theorem 1, the regularized NCP loss uniquely identifies the true operator, a guarantee that an alternative method encoding orthogonality directly into the model might fail to have. Namely, the orthogonality is defined via the true distributions and not the observed empirical ones. Thus, currently it is not clear how to properly encode it in the architecture and obtain the same level of theoretical guarantees.
>
> - __Q3:__ We added additional experiments both on synthetic and real high-dimensional data to illustrate that the NCP operator approach scales without any issue to high-dimensional, more complex data. Please see our global response for more details.

---

> > ### Comment · Reviewer_BaSA · 2024-08-12
> >
> > Thank you very much for a detailed reply! I confirm my positive view of this paper.

---

> > > ### Author Response · Authors · 2024-08-13
> > > **Acknowledgement to the reviewer**
> > >
> > > We would like to thank the reviewer for their comments, which inspired us to make additional steps and improve our work. We are happy that our rebuttal was helpful, and we commit to incorporate it in the revised manuscript.

---

### Official Review · Reviewer_n6sh · 2024-07-12

**Soundness:** 3
**Presentation:** 3
**Contribution:** 2
**Rating:** 6
**Confidence:** 2

**Summary:**

The paper proposes Neural Conditional Probability, a novel operator-theoretic approach to learning conditional probability distributions by learning parameters of the truncated SVD of the conditional expectation operator with a neural network. The authors provide a rigorous mathematical derivation and argue for statistical guarantees of their method. The empirical evaluations require major improvements to an otherwise solid paper.

**As a general note:** I do not consider myself an expert on the theoretical aspects of learning theory. My background is in Bayesian deep learning and simulation-based Bayesian inference. As such, my confidence regarding sections 3 and 5 is rather low, and my review shall mainly consult on the remaining sections that focus on presentation, embedding into other literature, and empirical evaluations.

**Strengths:**

- The introduction is excellent, with a high degree of accessibility for the broader NeurIPS community and sound motivation of the proposed method.
- The method seems mathematically rigorous, well-motivated, and sound.
- The authors compare their method against a high number of competing algorithms in the numerical experiments.

**Weaknesses:**

## Major
- The Related Work section does a good job of acknowledging related works that aim to learn conditional distributions. However, it utterly fails to embed the current paper into this research landscape. I recommend the authors elaborate on the precise similarities and differences between the referenced papers and their methods in the rebuttal period.
- The empirical evaluations are limited to low-dimensional toy problems. This is a stark contrast to the introduction of the method, where the authors repeatedly list the curse of dimensionality as a drawback of other methods. While I acknowledge that the paper is situated in the area of operator learning and ML theory, the quality standard of NeurIPS is not met by the authors’ experiments. This weak evaluation does not do the remainder of the paper justice and I strongly recommend the authors overhaul the experiments to feature high-dimensional tasks that cannot be solved with other state-of-the-art methods. This constitutes a major revision, and this is the main reason why I cannot recommend acceptance to NeurIPS 2024.


## Minor

- The empirical evaluation is missing some important information for real-world applications: What are the approximate wall-clock times for (1) training and (2) inference of the competing methods? Further, the authors mention the large required training set size, which might also influence the practically expected training duration in real-world tasks.
- Please fix the citations throughout the manuscript: Most citations are ‘text citations’ even if their embedding in the sentence warrants parenthesized citations (Author, 1976).
- This is just a personal preference, no need to address it: The ‘paper organization’ paragraph at the end of the introduction does not add value and the space could be used more efficiently elsewhere in the manuscript.
- The first sentence in the conclusion is incomplete.

**Questions:**

- As per your answer to checklist item 5 (Open access to data and code), I would like to request access to the full and reproducible code of the empirical evaluations.
- Since you want to compare your method with state-of-the-art conditional density estimation methods: Why don't you benchmark against conditional flow matching?

**Limitations:**

- The authors mention limitations throughout the manuscript, which I appreciate. However, I would recommend adding a dedicated **Limitations** section in the conclusion to give a compact overview for readers who don’t engage with the entire paper in detail.
- The performance of NCP in high dimensions might be a limitation, but the authors do not study this crucial setting.

---

> ### Author Rebuttal · Authors · 2024-08-07
>
> Thank you for your detailed and thoughtful feedback on our submission. We appreciate your recognition of our contribution to the field of operator learning and ML theory which was the primary objective of our work. We would like to address your concerns regarding the empirical evaluations and provide additional context.
>
> ## On general note
> Thank you very much for giving us the perspective to understand your review. While we followed all your suggestions for the revision, we would like to stress that theoretical guarantees developed in Sec. 3 and 5 are the integral part of this paper, and the heart of key contributions. Without being too technical, let us just briefly summarize what we feel as the most important aspects:
> - NCP is based on  __deep representation learning__ by training two DNNs (architecture depending on the data modalities) via a new loss so that the learned representations (u’s and v’s) can lead to __inference of diverse statistics derived from the conditional probability__.
> - For each of the considered statistics we __prove finite sample generalisation bounds in high probability__ that rely on the __concentration inequalities__ and __optimality gap from DNN training__.
> - To the best of our knowledge, this is __the first work__ to show how to __provably infer__ conditional probability distribution __using finitely many deep features__, so that the error bounds depend only on the effective problem dimension and  __not the ambient dimensions__.
>
> We hope that this summary brings more light to the quality and impact of our contributions.
>
> ## Weaknesses
>
> ### Major
> 1. We thank the reviewer for raising this point. __We added a discussion in the global response to situate our NCP approach in comparison to the four known strategies__. For the reviewer's convenience, we focus here on a detailed comparison with the best contenders to our NCP method, as shown in Tab. 1: Normalizing Flows (NF) and FlexCode (FC) from Izbicki and Lee (2017).
> FlexCode (FC) is an example of the direct learning strategy for conditional density estimation (CDE). Our results in Tab. 1 demonstrate that NCP, which learns an intrinsic representation adapted to the data, consistently outperforms FlexCode, which relies on a standard dictionary of functions, in all but one example. In the exception, NCP ties with FC. This also exemplifies that FC is not efficient for capturing the data's intrinsic structure when the FC model is overly misspecified. Regarding Normalizing Flows (NF), which is used in a conditional training strategy, a major limitation is the need to retrain an NF model for each conditioning. Additionally, NF builds a density model over the whole ambient space, which may not be efficient for capturing the data's intrinsic structure. Finally, to the best of our knowledge, NF lacks theoretical guarantees when applied to UQ tasks where strong guarantees of reliability are required. For more details on this aspect for the NCP, see our response to the general note above. We will incorporate this discussion in a revised version of our paper.
>
> 2. Thank you for this remark. __We added high-dimensional experiments focused on UQ tasks. While most are synthetic in order to explore, as requested, the scalability of NCP, one is a complex real-world problem of protein folding__, see also our global response. For the latter, we used a dataset that is considered as challenging for the task of inferring dynamics of the stochastic process governed by the conditional probability kernel. Applying NCP to this setting, we were able to infer this conditional density from learned GNN representations of Chignolin mini-protein ensemble. This allowed us to predict the expected average pairwise atomic distances between heavy atoms and infer its 10th and 90th percentiles. As reported in Fig. 3, folded/unfolded protein corresponds to small/large coverage that originates from less/more variations of the ensemble. To the best of our knowledge, this is the first work that was able to perform UQ to infer metastable states of Chignolin, an important contribution in its own right.
>
> ### Minor
> - We added an experiment on how time complexity and inference quality scales w.r.t. dimensionality of $X$. In the attached pdf we included Fig. 2 presenting compute times for training and statistical accuracy for conditional density estimation (CDE) using the KS-distance.
> - Thank you a nice suggestion, we will follow it.
>
> ## Questions
> 1. As requested, __we provided the AC with a link to an anonymous repository__, as outlined in the NeurIPS guidelines, that includes notebooks for reproducing all our experiments.
>
> 2. We thank the referee for bringing this method to our attention. From our understanding, conditional flow matching is designed to efficiently train continuous normalizing flows in high-dimensional settings, particularly for image generation tasks. However, based on our review of the relevant literature, this method has not been explored for the purpose of UQ. In our benchmark, we used a discrete-time masked autoregressive normalizing flow, which performs well on our synthetic benchmark due to its universal approximation property [Papamakarios et al. Normalizing Flows for Probabilistic Modeling and Inference. JMLR 2021, 22(57):1−64]. We did not encounter any issues with training convergence for this model. Additionally, our primary objective is to provide a simple yet effective approach for the UQ tasks we investigated. For these reasons, we believe that incorporating this more complex model in our benchmark lies outside the scope of our study.
>
> ## Limitations
> As elaborated above, we added additional experiments both on synthetic and real data to __show that the NCP operator approach doesn't suffer from this limitation__. Indeed it scales without any issue to high-dimensional and more complex data, without degradation of inference quality. Lastly, as requested, we will collect all discussions regarding limitations in one paragraph.

---

> > ### Comment · Reviewer_n6sh · 2024-08-08
> >
> > Thank you for the detailed rebuttal which addresses my main concerns regarding the empirical evaluations. I acknowledge that the theoretical aspects of the paper are the main contribution. I will raise my score accordingly.
> >
> >
> > ## Flow matching and UQ
> >
> > > From our understanding, conditional flow matching is designed to efficiently train continuous normalizing flows [...] However, based on our review of the relevant literature, this method has not been explored for the purpose of UQ.
> >
> > Flow matching has been studied for simulation-based inference, which is concerned with the very question of uncertainty quantification. See reference [1] which also contains a code repository.
> >
> > [1] Dax et al. (2023). Flow Matching for Scalable Simulation-Based Inference. NeurIPS 2023. https://arxiv.org/abs/2305.17161

---

> > > ### Author Response · Authors · 2024-08-13
> > > **Acknowledgement to the reviewer**
> > >
> > > We would like to thank the reviewer for their comments, which inspired us to make additional steps and improve our work. We are happy that our rebuttal was helpful, and we commit to incorporate it in the revised manuscript.
> > >
> > > Concerning the reference, we thank reviewer for bringing this paper to our attention, we will definitely include it in the revision.

---

### Author Rebuttal · Authors · 2024-08-06

We thank all reviewers for their insightful evaluation of our paper. We appreciate all their comments and remarks, which we will incorporate in our revision. Before addressing each review in detail, we would like to point out some general remarks that apply to all of them.

## Positioning

Our main focus is on advancing Uncertainty Quantification (UQ) to handle nonlinear structures effectively. Several high-stakes nonlinear conditional probability estimation problems, such as CVAR for financial regulators (Since the 2008 financial crisis, regulators have required banks to limit their risk exposure, necessitating accurate computation of conditional value at risk, which relates to the conditional CDF), meaningful uncertainty quantification for engineering design (e.g., to drastically reduce the number of expensive real-world prototypes and tests in favour of a more computationally driven process), and rare-events prediction for predictive maintenance, are typically formulated in state spaces of moderate to high dimensions and often involve large datasets with complex, nonlinear structures, such as data residing on manifolds or graph data. Our goal is to showcase that the NCP method can effectively handle such structured data, offering a flexible and scalable modelling general solution. Importantly, we provide solid and provable statistical guarantees, aiming to advance safe and secure AI applications.

## Contributions

NCP leverages operator theory, a rich and powerful area of mathematics, allowing us to gain deep insights into the structure of functional spaces including probability distribution spaces. To the best of our knowledge, the NCP approach integrating operator theory in a self-supervised learning framework represents deeply innovative idea with significant potential for a range of ambitious applications. In the current paper, we focus on presenting NCP in the simplest possible way to illustrate its core benefits.

**Strengths of NCP**

1. **Versatility:** NCP combined with a simple and compact MLP architecture, consistently outperforms more complex models and, when it does not, reliably matches their performance.

2. **Adaptation to Intrinsic Dimension:** NCP effectively adapts to the intrinsic dimension of the data, constructing a well-suited low-dimensional representation space without requiring prior knowledge. This results in a more efficient handling of the data's inherent structure.

3. **Efficiency in Training:** NCP involves only a single training phase for the joint density. Once trained, we can easily derive all conditional probabilities without additional retraining, streamlining the process. In addition, NCP is easy and fast to train. Indeed we prove that it also scales linearly with latent space dimension and sample size. Our new experiments confirm that NCP scales with the ambient dimension, with compute time increasing at most sublinearly with the dimension (See Fig 2).

Additionally, our method also feature the following theoretical and conceptual strengths:

4. **Complete Theory:** Our approach is supported by a robust theoretical framework, providing strong guarantees for performance and reliability in both training and UQ accuracy.

5. **Assumption Comparison:** The density assumption with respect to marginals for NCP is significantly weaker compared to many other methods like Normalizing Flows (NF), which usually require density with respect to Lebesgue measure. This makes our approach broadly applicable, for instance for discrete distributions, or complex structured data like manifolds.


## Situating NCP method in the landscape of CDE methods

While NCP does not strictly fall into any of the four categories described by Gao and Hastie (2022), it is perhaps closest to the direct learning strategy but represents a significant improvement. Unlike the direct learning strategy, which uses a pre-specified kernel or a dictionary of functions to approximate the target conditional density, NCP directly learns the joint density by finding a low-dimensional latent space representation that efficiently adapts to the data. This is illustrated in our experiments where NCP demonstrated its versatility in handling diverse data including manifolds, graphs, continuous and discrete distributions (see Fig. 1, 2 and 3 in the attached pdf). Moreover, by learning proper representation adapted to the data, NCP can capture the intrinsic dimension of the data, which is reflected in our learning guarantees that depend on the latent space dimension rather than the ambient space dimension, even in the absence of prior knowledge. This theoretical foundation and added experiments (see our next point) demonstrate that NCP is capable of handling high-dimensional data.

## Challenging Experiments

As requested, we conducted additional experiments to further demonstrate the robustness and versatility of our approach across different dimensionalities and data structures. We have reported the results in the rebuttal pdf file and provided the AC with a link to an anonymous repository containing all code necessary to rerun our experiments.

We conducted high-dimensional experiments on synthetic data with discrete and continuous distributions scenarios, where the joint distribution is governed by a hidden low-dimensional manifold in a higher-dimensional space. Results in Fig. 1 and 2 in the attached pdf include KS-distance and computation times across various dimensions and conditional CDF plots for continuous and discrete distributions parameterised by a manifold. Our comparison of NCP and NF shows that NCP performs well and scales effectively in high-dimensional settings.

Additionally, an experiment on a real-world challenging application on folding dynamics of a protein is performed. There NCP is combined with a graph neural net architecture to forecast the quantiles of a physically relevant observable and infer the change from folded to unfolded state, see Fig. 3 for details.

---

### Comment · Area_Chair_kzAj · 2024-08-12
**Code now available**

Hello reviewers,

There has been some discussion of wanting to examine the code for this paper. The authors (following the official guidelines) have sent me a link to a repo containing the code. I have checked, to the best of my ability, that this is anonymous, so I am passing on the link to you here: https://anonymous.4open.science/r/NCP/README.md

---

### Decision · Program_Chairs · 2024-09-25

**Decision:**

Accept (poster)

**Comment:**

Intuitively, the idea of this paper is very simple—essentially: Learn neural networks that transform x and y to some feature space, so that the joint density is the inner-product of the feature spaces. This has an obvious problem, namely that the result may not be normalized (or even positive). Seemingly for this reason (though the issue is never really acknowledged in the paper) a quadratic loss is used that essentially measures the squared difference between the learned "density" (not guaranteed to be normalized or positive) and the target. Then, an extra regularizer is introduced that encourages these feature spaces to be orthogonal. A high-probability bound result is given.

Reviewers were uniformly in favor of accepting this paper. The method appears to be novel and the results correct. But I do have some concerns that the presentation greatly obscures the simplicity of the underlying method. It seems perverse to not first describe the method in simple terms before giving the measure-theoretic motivation. (Which, after all, is just a motivation.) The theory makes quite strong assumptions (as is basically necessary given the presence of neural networks inside the method. As a service to the community, I urge the authors to be more clear about the simplicity of what is being proposed, and not to "hide" the simplicity of what is actually being proposed behind so much measure theory.